# Are heavy rainfall events a major trigger of associated natural hazards along the German rail network?

Sonja Szymczak[1], Frederick Bott[1], Vigile Marie Fabella[1], Katharina Fricke[1]

[1]German Centre for Rail Traffic Research at the Federal Railway Authority, Dresden, 01219, Germany

*Correspondence to*: Sonja Szymczak (SzymczakS@dzsf.bund.de)

**Abstract.** Heavy rainfall events and associated natural hazards pose a major threat to rail transport and infrastructure. In this study, the correlation between heavy rainfall events and three associated natural hazards were investigated using GIS analyses and random-effects logistic models. The spatio-temporal linkage of a damage database of DB Netz AG and the CatRaRE-catalogue of the German Weather Service revealed that almost every part of the German rail network was affected by at least

one heavy rainfall event between 2011–2021. Twenty-three percent of the flood events, 14 % of the gravitational mass movements and 2 % of the tree fall events occurred after a heavy rainfall event. The random effects logistic regression models showed that a heavy rainfall event significantly increases the odds of occurrence of a flood (tree fall) by a factor of 22.7 (3.62), respectively. We find no evidence of an effect for gravitational mass movements. The heavy rainfall index and the 21-days antecedent precipitation index were determined as characteristics of the heavy rainfall events with the strongest impact on all

three natural hazards. The results underline the importance of gaining more precise knowledge about the impact of climate triggers on natural hazard-related disturbances in order to make rail transport more resilient.

## 1 Introduction

Heavy rainfall events are one of the most important triggers for flash floods, which can have catastrophic effects on the affected

regions. A prominent recent example is the flood disaster in Western Europe in July 2021 with over 200 fatalities (Kreienkamp et al., 2021). During the period 12 to 15 July 2021 extreme rainfall occurred in Germany and the Benelux countries (Junghänel et al., 2021; Tradowsky et al., 2023). The resulting flash floods caused considerable damage to infrastructure such as houses (Korswagen et al., 2022), communication facilities, roads and railway lines (Szymczak et al., 2022), making the event the deadliest European flooding event in nearly three decades and the costliest on record (Aon, 2021). Damage to critical

infrastructures such as power supply and transportation is of particular concern, as efficient infrastructure is important to ensure that affected regions can be reached and supplied with essential goods even in the event of a disaster.

Fortunately, not every heavy rainfall event has such catastrophic effects as the example of July 2021. Nevertheless, at the local level, secondary processes triggered by heavy rainfall, such as landslides, flooding and scouring, can cause significant

economic damage (e.g. Kjekstad and Highland, 2009; Lehmkuhl and Stauch, 2022), especially when transport infrastructure is affected (Klose et al., 2014; Winter et al., 2016). If such events occur along transport networks and disrupt traffic and transport, they are documented by the infrastructure operators. However, these damage databases rarely establish a cause-effect relationship, i.e. there is usually no precise information on which climatic or other parameter triggered the damaging event. This is because *reactive* natural hazard management, i.e. damage repair and rapid restoration of operations, is a higher

priority for operators than a detailed documentation of the triggering event. Nevertheless, it should not be neglected that a *proactive* approach, which includes a detailed analysis of the cause-effect relationship between climatic triggers and resulting natural events, contributes significantly to increasing the long-term resilience of transport infrastructure to natural hazards.

Within the framework of proactive natural hazard management, it is possible to identify regions that are particularly at risk,

e.g. by developing hazard indication maps, or to determine climatic thresholds for the triggering of certain processes. Particularly in view of the current climate change situation, the management of climatically induced natural hazards is becoming increasingly important in the transportation sector (Koks et al., 2019). Which natural hazards are particularly relevant depends on the region and the mode of transport. In addition to the climatic conditions of the respective region, special features specific to the mode of transport must also be considered. For example, line closures in rail transport have a

significantly higher impact than in road transport due to the lower number of alternative routes, and short-term bypasses of rail lines are associated with a higher logistical and personnel effort (Rachoy and Scheikl, 2006). Likewise, the risk of damage is higher due to the more complex infrastructure, rail-bound driving, longer braking distance and train length (Mattson and Jenelius, 2015).

In German railroad operations, tree falls, gravitational mass movements and flood events are particularly common natural hazards that cause operational disruptions (Fabella and Szymczak, 2022). These events can be triggered by a variety or a combination of different factors, but heavy rainfall events are possible triggers for all of these processes, as could be observed for example during the event in July 2021. As an increase in the intensity of daily and especially sub-daily extremes can be expected in a warmer climate (e.g. Lengfeld et al., 2021; Zeder and Fischer, 2022), special attention of transport operators

should be paid to precipitation extremes and associated hazards. In our study, we investigate the relationship between heavy rainfall events and associated natural hazards, such as floods, gravitational mass movements and tree falls, and its impact on the German wide rail network. For this purpose, we first perform a spatio-temporal linkage of a damage database of DB Netz AG (part of Deutsche Bahn, Germany's largest railroad company) and the catalogue of radar-based heavy rainfall events (CatRaRE) from the German Weather Service (DWD). This analysis should bring any spatial or temporal bias of the heavy

rainfall events and the investigated natural hazards to light. Secondly, we set up random-effects logistic regression models to explore (1) whether the odds of the occurrence of natural hazards increase significantly with proximity to a heavy rainfall event and (2) which characteristics of the heavy rainfall events have the strongest impact on the occurrence of the natural hazards.

The logistic regression, although customarily used in data science as a forecasting tool, was used in this study in order to fully explore and shed light on the nuances of the complex relationship between heavy rainfall and natural hazards.


## 2 Materials and Methods

### 2.1 Datasets

#### 2.1.1 CatRaRE catalogue of the German Weather Service (DWD)

For Germany, the DWD has developed the so called CatRaRE catalogue, a catalogue of heavy rainfall events collected via radar to provide a comprehensive overview on all heavy rainfall events that have occurred in Germany since 2001 (Lengfeld et al., 2021). Each event is described by various parameters such as time, duration, location, mean and maximum precipitation, severity indices as well as meteorological, geographical and demographic information. Strictly speaking, the CatRaRE catalogue consists of two catalogues: T5 and W3 (Lengfeld et al., 2021). As no standardized guideline for defining heavy

rainfall exists, events for the catalogue were extracted by either (1) their intensity with Warning Level W3 (events with 25-40 l/m² in 1 hour or 35-60 l/m² in 6 hours) of the official DWD warning levels used as a threshold (W3-catalogue) or (2) their return period taking local conditions into account (T5-catalogue). We decided to use the W3-catalogue for our analysis as it is more suitable for Germany-wide studies because of the uniform threshold for heavy rainfall events (Lengfeld et al., 2021). As event data from the database of DB Netz AG is only available for the years 2011-2021, only heavy rainfall events from these

years were included in our analysis. A total number of 14275 heavy rainfall events occurred in these 11 years. Not all of these events are relevant for our study, since only 7722 events can be spatially intersected with the German rail network. Throughout the study period, the proportion of events that can be spatially intersected with the rail network remains constant per year at around 50%. The largest number of events affecting the rail network occurred in 2018 (1160), the lowest in 2012 (454) (Figure 1a). According to Lengfeld et al. (2021), 2018 belongs to the years with the highest number of heavy rainfall events over the

entire observation period (2001-2021). The monthly distribution shows a clear seasonal pattern with the majority of events (5682, 73.6%) occurring in summer (JJA, see Figure 1b). In addition, many events occurred in May and September while heavy rainfall events were rare during winter. This is consistent with the distribution over the entire period 2001-2021, as May-August are the most eventful months here (Lengfeld et al. 2021).

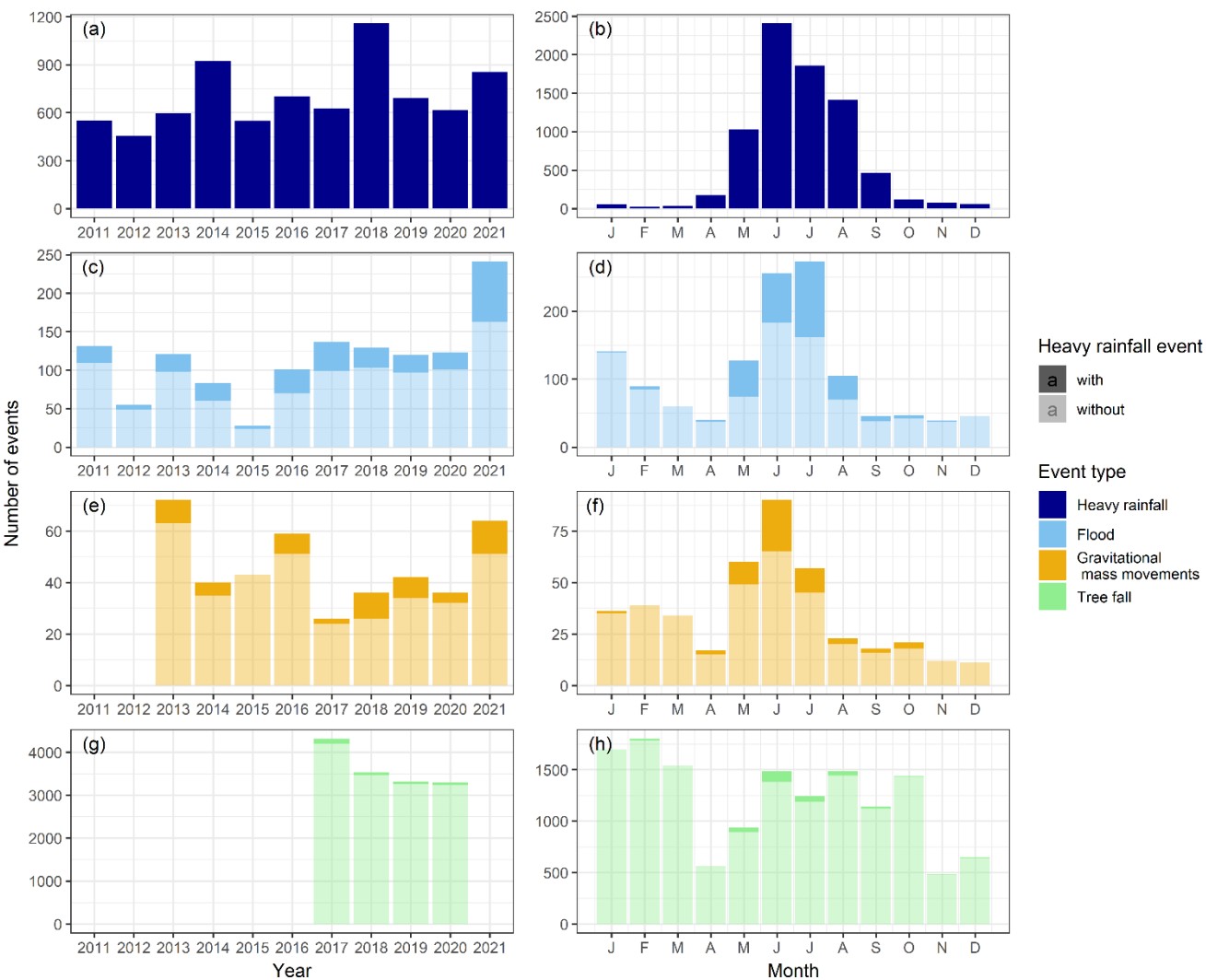

Figure 1: Monthly and yearly distribution of heavy rainfall events (Datasource: CatRaRE catalogue) spatially intersected with the German rail network, and gravitational mass movement, flood and tree fall events along the German rail network recorded by the damage database of DB Netz AG. The darker areas of the bars (c – h) include the events where a heavy rainfall event occurred up to two days prior to the event.

The spatial distribution of all heavy rainfall events spatially intersected with the German rail network is shown in Figure 2. The spatial reference used for this analysis were track sections as defined by the GIS-layer "geo-strecke" provided by DB Netz AG, resulting in a total of 15939 track sections. The events are distributed over all regions of Germany with a focus in southern Germany (federal states of Bavaria and Baden-Wuerttemberg). Over the 11-year period, there are very few track sections (437), which were not affected by at least one heavy rainfall event, while most of the pre-alpine railway lines in southern Germany were affected by more than 30 events. However, the "Starkregenindex" SRI, an index describing the speed at which

rainfall accumulates within a specified duration of time, of these events is in general lower. Highest mean SRI-values are recorded in the northern part of Germany, mainly in the federal state of Lower Saxony.

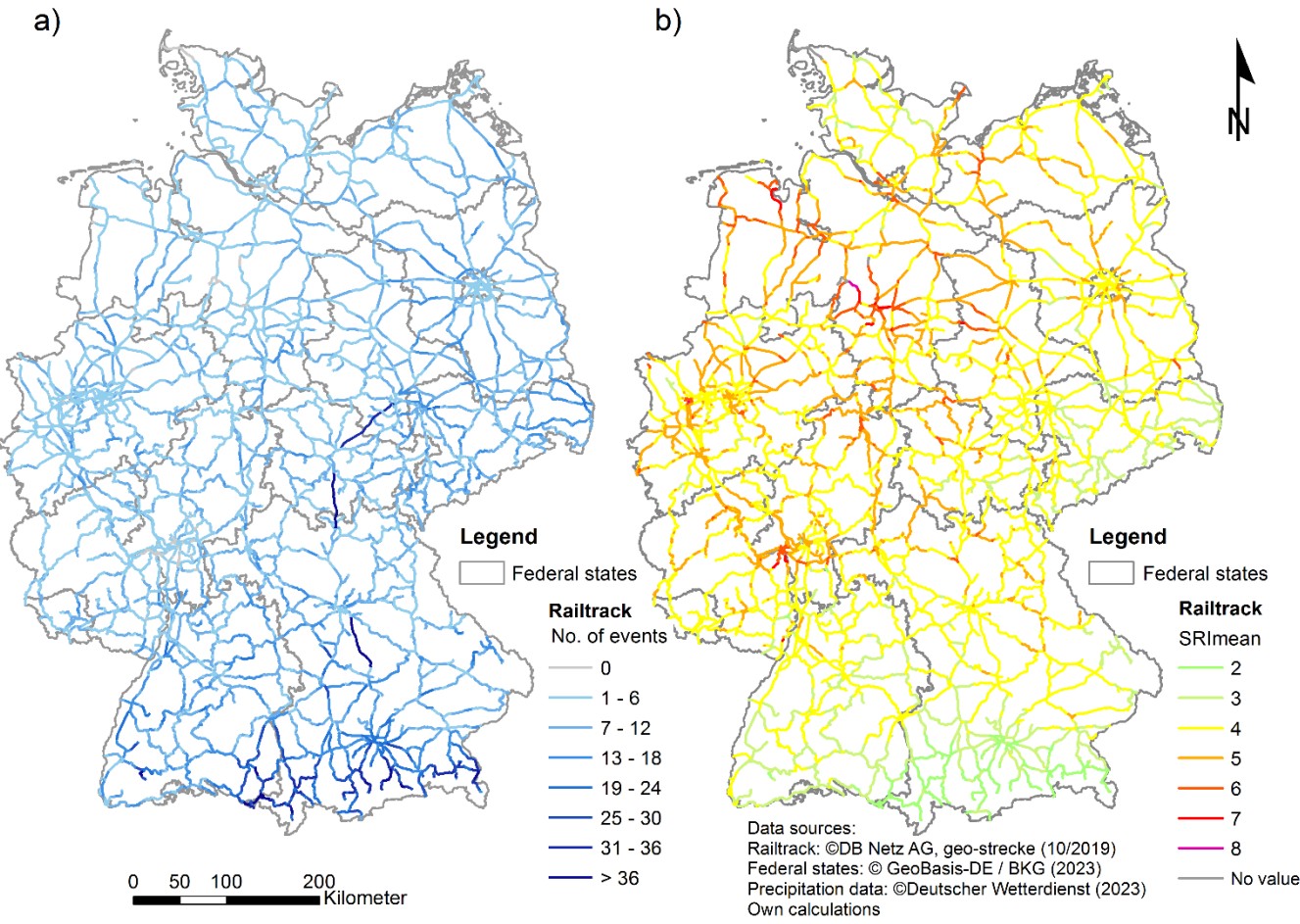

**Figure 2: Spatial intersection of heavy rainfall events from the CatRaRE catalogue and the German rail network for the time period 2011-2021. a) Number of events per track section. b) Mean SRI-values ("Starkregenindex", for definition refer to Table 1) for all events per track section. The SRI is calculated for every heavy rainfall event and ranges from 0-12. Note that in this figure mean values for several events are shown, limiting the resulting SRI-values to the range 2-8. Data sources: "geo-strecke" 10/2019 DB Netz AG (rail network), GeoBasis-DE / BKG 2023 (federal states), Deutscher Wetterdienst (heavy rainfall events).**

### 2.1.2 Damage database for the German rail network

The event data of the natural hazards along the German rail network were extracted from a damage database of DB Netz AG. In the database, each disruption along the rail infrastructure is documented with a time stamp, the event location and a short event description. As this database and data collection is not restricted to natural hazard-specific incidents, the events relevant to this study were filtered using an extended text search with event-specific search terms and then checked manually for correctness and double notification. This procedure cannot verify that all events were actually extracted from the database ('completeness') and that there are no false negatives, as the textual descriptions do not follow a fully consistent categorization

and thus not all keywords may have been correctly identified. However, the two-step extraction with subsequent manual control of the data ensures the 'correctness' of the data insofar as there are no false positives and no events are contained in the database export due to incorrect assignment. The distribution of false negatives is assumed to be fairly even throughout the study period due to the invariant methods of data collection and filtering.


In the following, the three resulting sub-databases for flood events, gravitational mass movements and tree falls are briefly described. The flood dataset contains in total 1269 events for the period 1 January 2011-31 December 2021, which include, but are not further categorized into river floods or local flash floods. The most eventful years were 2021 (241), 2017 (137), 2011 (131) and 2018 (129), while the least eventful years were 2012 (55) and 2015 (28) (Figure 1c). Flood events occurred
mainly between May and August with a high concentration in June and July, but also in January (Figure 1d). In contrast, they were rare between September and December. The gravitational mass movement dataset includes a total of 418 events for the period 1 January 2013-31 December 2021, with the most eventful years being 2013 (72), 2021 (64) and 2016 (59), and the least eventful being 2018 and 2020 (36 each) and 2017 (26) (Figure 1e). The monthly distribution showed a concentration of events between May and July and a second, smaller peak between January and March (Figure 1f). The tree fall dataset includes
a total of 14461 events for the period 1 January 2017-16 December 2020. The most eventful year was 2017 (4319), the least eventful 2020 (3301) (Figure 1g). However, as the last 15 days of the year are missing in 2020, it is also possible that 2019 is the least eventful year (3310). The seasonal distribution of tree fall events is not as pronounced as for the other two processes. Tree fall events occurred mainly between January and March as well as between June and October (Figure 1h).

### 2.1.3 Explanatory control variables

Additional climatological and hydrometeorological variables related to the investigated natural hazards were used to serve as explanatory control variables and to check for other relationships in the statistical regression analysis. These variables were derived from publicly available datasets provided by the DWD. Daily precipitation values were used from gridded observational datasets of precipitation provided by the HYRAS dataset (Razafimaharo et al., 2020). This dataset is based on precipitation measurements for Germany and its neighboring countries and interpolates them into 5 km x 5 km grids, taking
into account topographic and other effects. Daily values of soil moisture were used from a 1 km x 1 km grid developed by the DWD for agrometeorological applications. These values are interpolated from soil moisture in 60 cm depth under grass at a fixed selection of stations (Löpmeier, 1994). Also included was the hazard indication map for slope and embankment landslides along the German rail tracks provided by the German Centre for Rail Traffic Research at the Federal Railway Authority, which is modeled based on the geology, morphology and land use characteristics of the area surrounding the rail tracks (Kallmeier et
al., 2018).

## 2.2 Methods

### 2.2.1 Intersection of heavy rainfall events with events from the damage database

The analysis of the spatial and temporal relationship between the heavy rainfall events and damage events along the German rail network was carried out by intersecting the CatRaRE polygon data provided by the DWD and the compiled railway damage database. First, the spatial intersection was carried out using the GIS software ArcMap, version 10.8.1. In ArcMap, the respective damage events floods, gravitational mass movements and tree falls, which are available as point information, were intersected with the CatRaRE heavy rainfall events (W3-catalogue) between 2011 and 2021, which are available as area polygons, using the tool "Spatial Join". In the process multiple join features (heavy rainfall events) were assigned to each target feature (damage event ("Join one to many")). This creates a database in which all spatially overlapping heavy rainfall events are assigned to the damage events. Thus, there are event locations where more than 50 heavy rainfall events from 2011 to 2021 can be found.

A heavy rainfall event can only be considered as a trigger for a damage event if the heavy rainfall event occurs directly or shortly before the damage event. As a heavy rainfall event usually is an event of short duration and high intensity, in general the time lag between trigger and effect is rather short (e.g. shown for shallow landslides by Zêzere et al. (2015) and for landslides during summer by Rupp (2022)). However, heavy rainfall events often occur during weather conditions that lead to clusters of rainfall events, so that the occurrence of several heavy rainfall events in succession can also be a possible cause (e.g. shown for deep landslides by Bevacqua et al. (2021) and for tree fall by Locosselli et al. (2021)). As there is no generally accepted threshold, we have chosen in our study to consider all heavy rainfall events that occurred up to two days before the damage event. This considers possible inaccuracies in the DB damage database, as the date in the damage database represents the time when the event was recorded. This does not necessarily coincide with the actual occurrence of the event, as, for example, events that occur at night are often not recorded until the following day during the first train journey of the day. Furthermore, the selected time period was supported by an analysis of the natural breaks in the data set. The selection of the events was conducted by temporal intersection using the function "DateDiff" in ArcMap. Since both the damage events and the heavy rainfall events have a day-accurate time stamp, the difference in days between the start of the heavy rainfall event and the occurrence of the damage event could be identified.

### 2.2.2 Extraction of explanatory control variables

The corresponding values from the explanatory control variables daily precipitation, daily soil moisture and hazard class of landslide risk were extracted from the gridded data at the location and, when applicable, for the date of the event occurrence using the python libraries gdal and ogr.

### 2.2.3 Statistical analysis and modelling

For the statistical investigation, a panel data analysis as well as a cross-sectional analysis was carried out. The panel data analysis was conducted to test whether the odds of the occurrence of natural hazards is affected by a heavy rainfall event, and whether the odds increases with proximity to a heavy rainfall event. Panel data allows to consider observations over several points in time, which is crucial for measuring the temporal proximity to a heavy rainfall event at a route segment. Therefore, it is possible to compare the effects of heavy rainfall events that occur at different times before a natural hazard event, e.g. two days before, one day before or at the same day. The cross-sectional analysis was conducted to examine which characteristics of a heavy rainfall event have the strongest effect on natural hazard occurrence. In cross-sectional analyses, each observation is only considered at a single point in time.

#### 2.2.3.1 Panel data analysis

For the panel data analysis, the dataset was created with route segments as the cross-sectional unit and day as the time-series unit. A route segment is defined as a section of the German rail network between two adjacent operating points. The total length of the German rail network owned by DB is 56939 km of tracks and was divided into 9679 route segments for our dataset. The segments differ in length between 140 m and 12.7 km with an average length of 3.4 km. Route segments were chosen as the cross-sectional unit as it is on the one hand the smallest operational unit used by DB that can represent the complete rail network. On the other hand, the number of route segments still allows for a tractable data size that does not inflate the calculation times in the statistical analysis, for example compared to taking 5-meter segments across the entire network. The period under consideration were the years between 2011 and 2021 for each route segment, for which it must be tested whether a heavy rainfall event has occurred or not. To calculate 30-days antecedent precipitation (one of the control variables) for each day and route segment, the period started with 1 February 2011, so that the complete dataset is available for 3987 days (= time-series units), resulting in a total of 38590173 route segment - day combinations, hereafter referenced as observations. The number of observations used in the succeeding models vary depending on the available time period of the natural hazard event datasets.

Each observation was spatially intersected with the CatRaRE catalogue and the explanatory control variables based on the coordinates of the segment's starting point. The segment is considered to have been affected by a heavy rainfall event on a given day if a heavy rainfall event from the CatRaRE database has occurred on that day up to a maximum of two days previously. This is then indicated by a binary variable. The flood, gravitational mass movement and tree fall events from the DB damage database were matched to route segments based on their reported route number and kilometer. A natural hazard event can affect more than one route segment. A binary variable was then created for each natural hazard event, which takes the value of one if the respective event was reported on the route segment on that day and zero otherwise.

To test if the odds of the occurrence of natural hazards increase with proximity to a heavy rainfall event, a random-effects logistic regression (logit) model was used. Although the logit approach is conventionally used with the aim of forecasting, it can also be applied to questions of inference, as in this case, where it is used to elucidate the effect of heavy rain on the occurrence of natural hazards. Taking $p = Pr(Y = 1)$ to be the probability that a natural hazard event occurs (where $Y$ is either a flood, gravitational mass movement or tree fall), the relationship between this probability $p$ and a heavy rainfall event ($HR$) was modeled using a logit link function, such that

$$logit(p) = ln\left(\frac{p}{1-p}\right) = \beta_0 + \beta_1 HR + \boldsymbol{\beta'_2 x} + \boldsymbol{\beta'_3 z} + \boldsymbol{\beta'_4}(HR * \boldsymbol{x}) \tag{1}$$

where $\boldsymbol{x}$ is a vector of explanatory control variables, $\boldsymbol{z}$ is a vector of seasonal and yearly dummies, and ($HR * \boldsymbol{x}$) is the interaction between heavy rainfall and the control variables. The parameters $\beta_0$, $\beta_1$, $\boldsymbol{\beta'_{2_2}}$, $\boldsymbol{\beta'_3}$, and $\boldsymbol{\beta'_4}$ are the corresponding scalar and vector coefficients. The vector of controls $\boldsymbol{x}$ contain the following variables

$$\boldsymbol{x} = \begin{bmatrix} \text{Daily precipitation} \\ 30 - \text{Day accumulated precipitation} \\ \text{Daily soil moisture} \\ (\text{Daily soil moisture})^2 \end{bmatrix}$$

To account for the potential non-linear effect of soil moisture on the incidence of natural hazards due to the non-linear relationship between soil water content and soil matric potential (Rawls, et al. 1993; Zhu, et al. 2022; Vichta, et al. 2024), we include the square of daily soil moisture in $\boldsymbol{x}$. Season and year dummies are included as the vector $\boldsymbol{z}$ to control for seasonal effects and effects caused by particular years with climactic extremes, as well as to account for the fact that the number of natural hazards varies greatly in different years and seasons. To test whether there are interaction effects between the control variables and heavy rainfall events, the following interaction terms are added to the equations in (1): daily precipitation $* HR$, 30-day accumulated precipitation $* HR$, daily soil moisture $* HR$, and (daily soil moisture)$^2 * HR$.

The logit function in equation (1) is simply the natural log of the odds, that is, the natural log of the probability that a natural hazard event occurs ($p$) divided by the probability that it does not occur ($1 - p$). The basis of interpretation of the model lies in its exponential form, which results in the odds on the left-hand side of the equation:

$$\frac{p}{1-p} = e^{\beta_0} \cdot e^{\beta_1 HR} \cdot e^{\boldsymbol{\beta'_2 x}} \cdot e^{\boldsymbol{\beta'_3 z}} \cdot e^{\boldsymbol{\beta'_4}(HR*\boldsymbol{x})}$$

Taking $HR$ to be a binary variable with a value of one when heavy rainfall occurred in the last two days and zero otherwise, then the odds ratio ($OR$) between heavy rainfall and no rainfall event becomes:

$$OR = \frac{\left(\frac{p}{1-p} \mid HR = 1\right)}{\left(\frac{p}{1-p} \mid HR = 0\right)} = e^{\beta_1} \cdot e^{\boldsymbol{\beta'_4 x}} \tag{1.1}$$

If indeed a heavy rainfall event increases the odds of a natural hazard event occurring, then the numerator of the odds ratio should be greater than the denominator, hence the odds ratio should exceed one. Note that the odds ratio will depend on the value of the control variables that are interacting with $HR$.

To test if the odds of a natural hazard event increases the closer it occurs to days with heavy rainfall events, a second logistic regression model similar to (1) was also estimated,

$$logit(p) = ln\left(\frac{p}{1-p}\right) = \beta_0 + \boldsymbol{\beta'_1}\,\boldsymbol{DHR} + \boldsymbol{\beta'_2}\boldsymbol{x} + \boldsymbol{\beta'_3}\boldsymbol{z} \tag{2}$$

where $\boldsymbol{DHR}$ takes the form of a vector of dummy variables representing the number of days after the heavy rainfall occurred,

$$\boldsymbol{DHR} = \begin{bmatrix} d_0 = \text{day of heavy rainfall} \\ d_1 = \text{one day after heavy rainfall} \\ d_2 = \text{two days after heavy rainfall} \end{bmatrix}$$

and $\boldsymbol{\beta'_1} = [\beta_{10}\quad \beta_{11}\quad \beta_{12}]$ are the corresponding parameter coefficients. Since interaction effects are already tested in the first model, interaction terms have been removed in this model for simplicity. The assumption that the odds of an event increases the closer it is in time to a heavy rainfall event is confirmed when the odds ratios follow the order $OR_0 > OR_1 > OR_2$, where

$$OR_j = \frac{\left(\frac{p}{1-p}\mid d_i\right)}{\left(\frac{p}{1-p}\mid d_{-1}\right)} = e^{\beta_{1j}}, \qquad j = 0, 1, 2.$$

with $d_{-1}$ as the reference category representing no heavy rainfall in the last two days.

Given the panel structure of the data, observations from the same route segment may be correlated with each other. To overcome this issue, models (1) and (2) were extended to include a random variable $\mu_i$ representing the unobserved individual heterogeneity of each route segment $i$. The final models are therefore

$$logit(p_{it}) = ln\left(\frac{p_{it}}{1-p_{it}}\right) = \beta_0 + \beta_1\,HR_{it} + \boldsymbol{\beta'_2}\boldsymbol{x_{it}} + \boldsymbol{\beta'_3}\boldsymbol{z_{it}} + \boldsymbol{\beta'_4}(HR * \boldsymbol{x_{it}}) + \mu_i \tag{3.1}$$

$$logit(p_{it}) = ln\left(\frac{p_{it}}{1-p_{it}}\right) = \beta_0 + \boldsymbol{\beta'_1}\,\boldsymbol{DHR_{it}} + \boldsymbol{\beta'_2}\boldsymbol{x_{it}} + \boldsymbol{\beta'_3}\boldsymbol{z_{it}} + \mu_i \tag{3.2}$$

where the subscript $t$ identifies the days in the sample, which differ based on the type of natural hazard (4011 days for floods, 3280 days for gravitational mass movement and 1461 days for tree fall). The parameters of the random-effects models are estimated using maximum likelihood. Given that all the explanatory variables in the models ($HR_{it}, \boldsymbol{DHR_{it}}$ and $\boldsymbol{x_{it}}$) are exogenous meteorological factors, the individual-specific component $\mu_i$ is expected to be uncorrelated with all the regressors in the models. The variable $\mu_i$ therefore represents the random effect for route segment $i$, which is typically assumed to be independently and identically distributed across route segments following a normal distribution $N(0, \sigma_\mu^2)$. Higher variance $\sigma_\mu^2$ indicates a higher correlation between two observations within the same route segment.

### 2.2.3.1 Cross-sectional analysis

Since heavy rainfall events differ considerably in intensity, duration and other features, a cross-sectional analysis was used to test which of these characteristics influence the occurrence of a natural hazard event. The cross-sectional dataset contains only those route segments hit by at least one heavy rainfall event between 2011 and 2021. This resulted in a total number of 9339 route segments, of which 8589 were affected more than once during the eleven-year period, on average about five times. Each combination of route segment and heavy rainfall event is considered as a separate observation in the cross-sectional dataset. From the panel data set, it can be determined whether a natural hazard event occurred during and up to two days after a heavy rainfall event on this specific route segment. For each heavy rainfall event, several characteristics are available in the CatRaRE catalogue, of which a selection was used in this study (Table 1).

Table 1: Abbreviations and descriptions of the characteristics of heavy rainfall events in the CatRaRE catalogue that were used for the analysis in this study.

| Abbreviation | Description |
| --- | --- |
| H | Duration [h] of the heavy rainfall event |
| RRmean | Mean precipitation [mm] of all RADKLIM pixels within the event zone |
| SRImean | Mean of the heavy rainfall index (in German "Starkregenindex"): An index describing the speed at which rainfall accumulates within a specified duration of time. Mean of all RADKLIM-pixels within the event zone (Range [0,12]) |
| V3_AVG | Mean of the 21-days antecedent precipitation index within the event zone |
| ETA | A measure of the extremity of the heavy rain event as a function of the return period as well as affected area of an event |
| VSGL_GRAD | Mean degree of sealing [%]: Percentage of sealed area including road infrastructure within the event zone |
| STRM_AVG | Mean elevation [m] above sea level within the event zone |
| TPI_AVG | Mean of the Topographic Position Index, 2 km circular neighborhood [m], in the event zone within Germany |

Considering a similar logistic model as in the panel analysis, the relationship between the characteristics of the heavy rainfall events and the probability $(p_i)$ that a natural hazard occurs in observation $i$, is assumed to take the form

$$logit(p_i) = ln\left(\frac{p_i}{1 - p_i}\right) = \beta_0 + \boldsymbol{\beta'_1}\boldsymbol{C_i} + \boldsymbol{\beta'_2}\boldsymbol{z_i} \qquad (4)$$

where $\boldsymbol{C_i} = [c_{i1} \quad \cdots \quad c_{i9}]$ is a vector of the eight aforementioned characteristics from the CatRaRE catalogue in Table 1, plus a variable for the slope and embankment landslide hazard index from the German Centre for Rail Traffic Research. The vector $\boldsymbol{z_i}$ is a vector of year and season control variables, and $\boldsymbol{\beta_1} = [\beta_{11} \quad \cdots \quad \beta_{19}]$ are the corresponding parameter coefficients. Since the variables in $\boldsymbol{C_i}$ are continuous, the interpretation of the odds ratios is based on a one-unit increase in the value of the variable of interest:

$$OR_j = \frac{\left(\frac{p}{1-p} \mid c_j + 1\right)}{\left(\frac{p}{1-p} \mid c_j\right)} = e^{\beta_{1j}}, \qquad j = 1, 2, \dots, 9.$$

The maximum likelihood method was used to estimate the parameters in this cross-sectional logistic model (4).

## 3 Results

### 3.1 Spatial intersection of heavy rainfall events and natural hazards

Of the 1269 flooding events, a total of 296 events (23 %) can be spatially and temporally linked to a heavy rainfall event. A total of 184 (62 %) of the flooding events linked to heavy rainfall occur in June and July with July being the front-runner (111 events) (Figure 1d). There are also a large number of coupled events in May and August, while the number is below ten events in the other months. The lowest number is in March and December (zero each) and January and November (two each). The distribution over the years varies between four (2015) and 78 (2021) events. Besides 2021, the most frequent overlaps occur

in 2016 and 2017. Of the 418 gravitational mass movement events, a total of 59 events (14 %) can be spatially and temporally linked to a heavy rainfall event, most of them (48 or 81 %) between May and July (Figure 1f). The distribution among the years varies between zero (2011, 2012, 2015) and 13 (2021) events. Besides 2021, the most frequent intersections occur in 2013, 2016, 2018 and 2019. Of the 14461 tree fall events, a total of 312 (2 %) events can be spatially and temporally linked to a heavy rainfall event. A total of 163 of the tree falls (35 %) linked to heavy rainfall occur in June and July with June being

the front-runner (108 events) (Figure 1h). There are also a large number of coupled events in May (40) and August (46), followed by September (21) and February (20). The lowest number occurs in November (1) and January (2). The distribution across years varies between 57 (2019) and 118 (2017) events.

    A comparative analysis of all three natural hazards shows that in all three processes mainly the hazard events in summer are

coupled with heavy rainfall events (Figure 3). In contrast, the hazard events in winter are predominantly not coupled with heavy rainfall events.

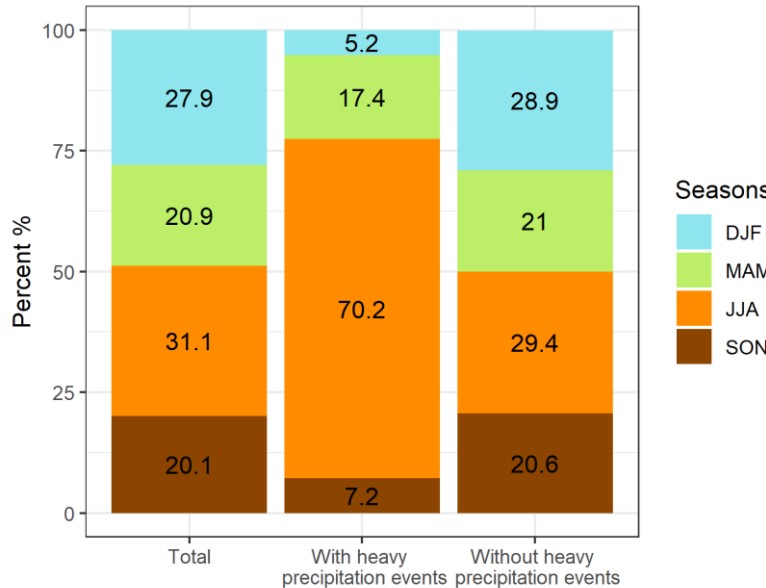

**Figure 3: Seasonal distribution of natural hazard events reported for the German rail network coupled with and without heavy rainfall events. All three natural hazard processes are shown together in the figure, as the distribution looks similar for each process when viewed individually.**

### 3.2 Influence of heavy rainfall events on the occurrence of natural hazard events

Table 2 provides the estimated odds ratios of the random effects logit models in equation (3) for the three different natural hazard events. The dataset of the entire period contains a total of 38590173 observations, but this number is lower for gravitational mass movements and tree falls because of the shorter available time period of the natural hazard event datasets. To evaluate model performance, several model criteria were calculated and presented in Table 2. Several values are provided to evaluate the goodness of fit for the models: The log likelihood is a function of the sample size, the higher the value the better. The rho value shows the contribution of the random effect to the total variance. The Akaike information criteria (AIC) is an estimator of the prediction error, the lower, the better a model fits the data it was generated from. The full regression tables with all explanatory variables and the evaluation of the model quality of the chosen model can be found in Appendix A. From this point forward, we will only be interpreting and discussing the results that are robust across the different models.

The exponentiated coefficients of heavy rain in Table 2 for the two hazards flood and tree fall are greater than one and statistically significant at 0.1 %. For gravitational mass movements, the coefficient is not statistically different from one. To further evaluate the magnitude of these effects, one must take into account the estimates of the interactions with the meteorological control variables. Since interaction terms are included in the model, the effect of heavy rain will depend on the level of precipitation, accumulated precipitation and daily soil moisture. In Table 3, we take the mean and median values of these meteorological control variables over the whole investigated time periods and calculate the odds ratio of heavy rain based

on equation (1.1). The large difference in magnitude between the main coefficient of heavy rain in Table 2 and the odds ratio of heavy rain in Table 3 particularly for tree fall and gravitational mass movements is primarily due to the exponentiated coefficient of the interaction between heavy rain and daily soil moisture. This coefficient is slightly greater than one for gravitational mass movements and less than one for tree fall. When raised to the power of the mean or median value of soil moisture, which is at least 75, the resulting number is very large for mass movements, and very small for tree fall. Following equation (1.1), this number is multiplied with the main coefficient, resulting in a substantial difference in the magnitudes.

**Table 2: Results of the random effects logit model for incidence of a natural hazard after a heavy rainfall event. The number of observations is lower for gravitational mass movements and tree fall events as for floods because of the shorter time period under consideration.**

|  | Dependent Variable | | |
|---|---|---|---|
|  | Flood | Gravitational Mass Movement | Tree Fall |
| Heavy rain, last 3 days=1 | **34.29**** | 3.812 | **39.85***** |
|  | (41.71) | (11.26) | (29.91) |
| Precipitation at route segment [mm] | **1.079***** | **1.052***** | **1.069***** |
|  | (0.00360) | (0.00691) | (0.00117) |
| Accumulated precipitation at route segment, 30 days [mm] | **1.010***** | **1.014***** | **1.003***** |
|  | (0.000843) | (0.00128) | (0.000293) |
| Daily soil moisture at route segment [% nFK] | **0.944***** | 0.957 | **0.931***** |
|  | (0.0138) | (0.0233) | (0.00316) |
| Daily soil moisture at route segment [% nFK], squared | **1.000***** | **1.000*** | **1.001***** |
|  | (0.0000939) | (0.000153) | (0.0000226) |
| Heavy rain, last 3 days=1 x Precipitation at route segment [mm] | **0.943***** | **0.956***** | **0.942***** |
|  | (0.00346) | (0.00792) | (0.00257) |
| Heavy rain, last 3 days=1 x Accumulated precipitation at route segment, 30 days [mm] | 0.999 | 0.996 | **0.997*** |
|  | (0.00125) | (0.00280) | (0.00137) |
| Heavy rain, last 3 days=1 x Daily soil moisture at route segment [% nFK] | 1.002 | 1.061 | **0.959*** |
|  | (0.0298) | (0.0742) | (0.0196) |
| Heavy rain, last 3 days=1 x Daily soil moisture at route segment [% nFK], squared | 0.9999 | 0.9995 | 1.0001 |
|  | (0.000179) | (0.000405) | (0.000128) |
| Observations | 38590173 | 31795515 | 14141019 |
| Number of route segments | 9679 | 9679 | 9679 |
| Log likelihood | -10645.3 | -4322.7 | -87853.1 |
| Rho | 0.430 | 0.531 | 0.375 |
| AIC | 21338.6 | 8689.5 | 175740.3 |

Exponentiated coefficients (odds ratios); Standard errors in parentheses; All models include season and year controls.
* $p < 0.05$, ** $p < 0.01$, *** $p < 0.001$

According to Table 3, when all meteorological control variables are at their mean (median) values, the odds of a flood event is on average 22.7 (25) times larger if a heavy rain occurred in the last two days than if no heavy rain occurred, respectively. The odds of a tree fall event on the other hand is on average 3.6 (4) times larger, when the meteorological factors are at their means (medians). For gravitational mass movements, the odds ratio is between 17 to 19 times larger, however, since the main effect of heavy rain on gravitational mass movement is not statistically significant, we will not interpret these values.

**Table 3: Odds ratios of heavy rain at the mean and median values of the precipitation, 30-day accumulated precipitation and daily soil moisture based on the estimates of the random effects logit model in Table 2; Odds ratios are calculated according to Eq. (1.1).**

| | *Dependent Variable* | | |
| --- | --- | --- | --- |
| | Flood | Gravitational Mass Movement | Tree Fall |
| Number of observations | 38590173 | 31795515 | 14141019 |
| Means | | | |
| - Precipitation | 1.926 | 1.922 | 1.856 |
| - 30-Day accumulated precipitation | 57.76 | 57.82 | 55.29 |
| - Daily soil moisture | 79.19 | 79.22 | 75.29 |
| **Odds ratio of heavy rain at the mean** | **22.70** | **17.90** | **3.616** |
| Medians | | | |
| - Precipitation | 0.100 | 0.100 | 0 |
| - 30-Day accumulated precipitation | 49.60 | 49.70 | 47.50 |
| - Daily soil moisture | 82 | 82 | 77 |
| **Odds ratio of heavy rain at the median** | **25.04** | **19.37** | **4.036** |

To provide insight in the temporal relationship between heavy rainfall events and resulting natural hazards, the random effects logit models were also calculated with the vector dummy variables in equation (3.2) representing the number of days after the heavy rainfall occurred (Table 4). Regarding the time lag, the odds of flood events is highest when the heavy rainfall event occurred on the same day as the flood event, and decreases with increasing temporal distance. All values are statistically significant. This means that compared to a situation with no occurring heavy rainfall, a heavy rainfall event is close to 12 times more likely to cause a flood on the same day, while 10 times more likely to cause a flood the day after, and almost 5 times more likely to cause a flood after two days.


For gravitational mass movement and tree fall events, the relationship is weaker than for flood events and even insignificant for heavy rainfall events occurring two days before the natural hazard event. Interestingly, the highest odds ratios can be observed for gravitational mass movements when the heavy rainfall event occurred one day before the natural hazard. In particular, the odds of a gravitational mass movement is close to eleven times higher one day after heavy rainfall compared to 350 a situation with no heavy rainfall, and more than three times higher on the day of heavy rainfall compared to no heavy rainfall. After two days, the odds of a gravitational mass movement is no longer different from a situation with no heavy rainfall. For tree fall events, the odds ratio on the day of a heavy rainfall is 0.333 and statistically significant, meaning that the odds of a tree fall event occurring on the same day as a heavy rainfall is less than a third that of a situation when no heavy rainfall occurs. In contrast, one day after a heavy rainfall event, a tree fall event is 2.4 times more likely to occur than in days with no heavy 355 rainfall. After two days, the odds ratio is no longer statistically different from one. A possible explanation for this observation is an operational one and lies in the way how data for tree fall is collected by Deutsche Bahn AG. Tree fall events are reported by train operators only upon encountering them en route. When a heavy storm or rainfall is expected, often train journeys are cancelled in advance to ensure the safety of passengers and employees. Therefore, less trains travel on days of heavy rain, making it less likely to encounter tree fall events on the same day. Most of the events are reported after the storm has settled.

In Tables 2 and 4, the odds ratios of the control variables precipitation and 30-day accumulated precipitation are statistically significant and slightly greater than one. The estimates are relatively smaller in magnitude compared to that of the heavy rainfall variables, which is to be expected from the continuous nature of the precipitation variables. In contrast, the results for daily soil moisture is ambiguous and not robust to changes in the specification of the heavy rainfall variable.


**Table 4: Results of the random effects logit model for incidence of a natural hazard with different numbers of days after a heavy rainfall event.**

| | *Dependent Variable* | | |
| --- | --- | --- | --- |
| | Flood | Gravitational Mass Movement | Tree Fall |
| Days from heavy rainfall event | | | |
|   -   day of heavy rainfall | **11.41**[***] | **3.584**[***] | **0.333**[***] |
| | (1.869) | (1.387) | (0.0525) |
|   -   1 day after heavy rainfall | **9.529**[***] | **10.58**[***] | **2.411**[***] |
| | (1.620) | (2.842) | (0.305) |
|   -   2 days after heavy rainfall | **4.757**[***] | 1.120 | 0.963 |
| | (1.082) | (0.810) | (0.195) |
| Precipitation at route segment [mm] | **1.023**[***] | **1.021**[***] | **1.053**[***] |
| | (0.00214) | (0.00466) | (0.00150) |
| Accumulated precipitation at route segment, 30 days [mm] | **1.011**[***] | **1.014**[***] | **1.004**[***] |
| | (0.000794) | (0.00127) | (0.000294) |
| Daily soil moisture at route segment [% nFK] | **0.955**[***] | 0.982 | **0.928**[***] |
| | (0.0120) | (0.0228) | (0.00316) |
| Daily soil moisture at route segment [% nFK], squared | **1.000**[***] | 1.000 | **1.001**[***] |
| | (0.0000791) | (0.000146) | (0.0000228) |
| Observations | 38590173 | 31795515 | 14141019 |
| Number of route segments | 9679 | 9679 | 9679 |
| Log likelihood | -10773.6 | -4343.0 | -88178.9 |
| Rho | 0.437 | 0.536 | 0.376 |
| AIC | 21591.2 | 8726.0 | 176387.9 |

Exponentiated coefficients (odds ratios); Standard errors in parentheses; All models include season and year controls and controls for landslide hazard.
$* p < 0.05, ** p < 0.01, *** p < 0.001$

The influence of the three control variables precipitation, accumulated precipitation of 30 days and daily soil moisture on the
relationship between heavy rainfall and the occurrence of a natural hazard event is depicted in Figure 4. Using predictive margins approach (Williams, 2012) and applying the regression models in Table 2, the predicted probability of a natural hazard event is calculated for each observation in the dataset for the case of no heavy rainfall and the case of a heavy rainfall event. The actual observed values of all the control variables were used to calculate the predicted probabilities. For observations with the same value of the meteorological variable, the average of the predicted probabilities were then taken. Therefore, for each
value of, say, precipitation, two points are obtained: the average probability with heavy rain (points on the dashed line) and the average probability without heavy rain (points on the solid line).

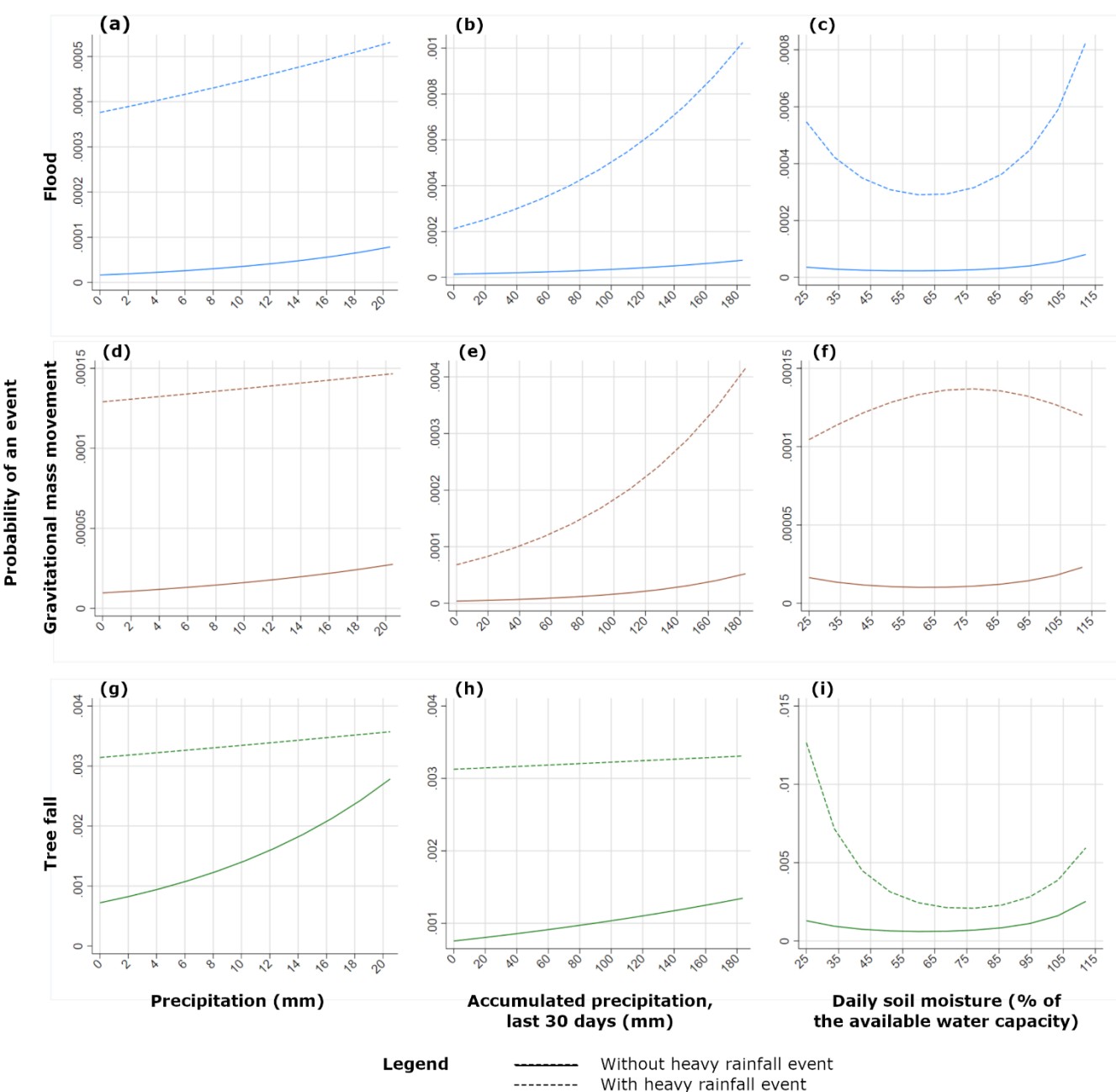

**Figure 4: The influence of the control variables precipitation, accumulated precipitation and soil moisture on the probability of occurrence of flood, gravitational mass movement and tree fall events. Each box compares the probability of occurrence for the two cases "without heavy rainfall event" and "with heavy rainfall event".**

The curves and the probability for the situation "with heavy rainfall event" are above the curves "without heavy rainfall event" for all three types of natural hazards and all three control variables, indicating that the probability of a natural hazard occurring

is always higher with a preceding heavy rainfall event. However, the curves have different shapes. In several subplots, both curves show a slight increase and the distance between them remains about the same (a), d), h)). This means that the difference in the probability of occurrence is independent of the amount of precipitation. In the case of b) and e), the distance becomes greater at higher values, i.e. the higher the amount of accumulated precipitation, the more a heavy rainfall event increases the probability of occurrence of a flood or gravitational mass movement. In the case of g), both curves slightly converge at high

values, i.e. the higher the amount of precipitation, the less a heavy rainfall event increases the probability of occurrence of a tree fall event.

For c) and i), the curve "with heavy rainfall event" has a U-shape. Thus, the probability of a natural hazard occurring during a heavy rainfall event is higher when the soil moisture takes on extreme values than when it takes on average values. There is evidence that extremely dry and extremely wet soil are determinants of floods (Vichta et al. 2024), mainly due to the

hydrophobic properties of soil, and oversaturation, respectively. Heavy rainfall in an environment with very dry and hydrophobic soil or very wet and oversaturated soil can therefore easily trigger a flood event, and this is reflected clearly in the results. In moderate soil moisture cases, where the soil can still absorb water brought about by heavy rains, the effect of heavy rain is then less pronounced. Similarly, it has been shown that drought stress can cause tree mortality (Grote, et al. 2016), meanwhile, soil oversaturation can cause waterlogging stress in trees (Gill, 1970; Kreuzweiser & Rennenberg, 2014).

With trees that are already under stress and vulnerable in very dry or very wet soil conditions, a heavy rainfall event could cause additional stress and be more likely to trigger tree fall. According to the data, the effect of heavy rain on tree fall is stronger for trees that are experiencing drought stress.

The arc shape in f) suggests that the probability of occurrence is highest at medium soil moisture values. However, since main effect of heavy rain on gravitational mass movement is insignificant, we will refrain from interpreting this result.

**3.3 Characteristics of heavy rainfall events and their influence on the occurrence of natural hazards**

The previous section has shown that the occurrence of heavy rainfall events has a statistically significant influence on the occurrence of natural hazards, particularly flood and tree fall events. However, as heavy rainfall events can be described with various parameters, the aim of the cross-sectional analysis was to investigate which characteristics of the heavy rainfall events affect the odds of natural hazards occurring and how these effects differ across the three processes. Table 5 presents the

resulting odds ratios of the estimated logistic regression model of the cross-sectional analysis when the parameter in question is increased by one unit. The duration of the heavy rainfall event and the mean precipitation throughout the area affected by the heavy rainfall event does not seem to have a significant effect on the odds of occurrence of a natural hazard. However, the heavy rainfall index (SRImean) does significantly increase the odds of all three natural hazards. When the index increases by one unit, the odds increase by a factor of 1.577 (floods), 1.716 (gravitational mass movements) and 1.389 (tree falls),

respectively. The table also reveals the significant effect of 21-days antecedent precipitation index (API) on all three types of

natural hazards. A one-millimeter increase in the API increases the odds by a factor of 1.055 (flood), 1.075 (gravitational mass movements) and 1.025 (tree fall), respectively.

The geographical characteristics within the heavy rainfall event zone that shows a significant influence on the occurrence of natural hazards are the degree of soil sealing and elevation. The degree of soil sealing has a negative effect on tree fall events and one percent of increased soil sealing reduces the odds by a factor of 0.936 (statistically significant at 0.1%). This could be due to the fact that more soil sealing means there are less trees in the area. The mean elevation within the heavy rainfall area reduces the odds of gravitational mass movement events by a factor of 0.998 (statistically significant at 1%). This unexpected observation may be due to the facts, that (1) the number of gravitational mass movement events in the available data set is very small and (2) the uneven distribution of railway lines in Germany in terms of elevation, with most lines being located in low-lying areas.

**Table 5: Results of the cross-sectional logit model on the components of heavy rainfall events and their effect on the odd ratios of the probability of occurrence of flood, gravitational mass movement and tree fall events. Note that the number of observations is reduced compared to Table 2 and 3, as the cross-sectional dataset contains only those route segments hit by at least one heavy rainfall event between 2011 and 2021.**

| | *Dependent Variable* | | |
| --- | --- | --- | --- |
| | Flood | Gravitational Mass Movement | Tree Fall |
| Duration of heavy rain [h] | 1.000 | 1.002 | 1.000 |
| | (0.002) | (0.005) | (0.002) |
| Mean precipitation [mm] of all pixels within the event zone (RRmean) | 1.015 | 1.007 | 1.002 |
| | (0.009) | (0.021) | (0.010) |
| Mean heavy precipitation index of all pixels within the event zone (SRImean) | **1.577**\*\*\* | **1.716**\*\*\* | **1.389**\*\*\* |
| | (0.103) | (0.242) | (0.089) |
| 21-days antecedent precipitation index - Mean within the event zone (V3_AVG) | **1.055**\*\*\* | **1.075**\*\*\* | **1.025**\*\* |
| | (0.008) | (0.017) | (0.009) |
| Extremity, mean throughout event duration (Eta) | 1.003 | 1.002 | 0.997 |
| | (0.002) | (0.005) | (0.002) |
| Degree of soil sealing [%] within the event area, mean (VSGL_GRAD) | 0.978 | 0.984 | **0.936**\*\*\* |
| | (0.013) | (0.021) | (0.014) |
| Mean elevation [m] above sea level in the event zone (STRM_AVG) | 1.000 | **0.998**\*\* | 0.999 |
| | (0.0003) | (0.001) | (0.0004) |
| Topographic Position Index [m] - Mean within event zone (TPI_AVG) | 1.049 | 1.016 | 0.982 |
| | (0.037) | (0.105) | (0.026) |
| Constant | 0.0001\*\*\* | 0.00001 | 0.0002 |
| | (0.0001) | (0.001) | (0.009) |
| Observations | 47605 | 41646 | 24132 |
| Log Likelihood | -1566.481 | -348.888 | -1326.230 |

| Akaike Inf. Crit. | 3180.963 | 741.777 | 2688.459 |

Exponentiated coefficients (odds ratios); Standard errors in parentheses; Season and year controls are included in all regressions.
[*] $p < 0.05$, [**] $p < 0.01$, [***] $p < 0.001$

## 4 Discussion

### 4.1 Heavy rainfall events and associated natural hazards

The heavy rainfall event in July 2021 was an exceptional event in terms of intensity and spatial extent (Tradowsky et al., 2023). Such devastating flash floods are therefore not to be expected with every heavy rainfall event occurring in Germany. Nevertheless, less intense heavy rainfall events are not a rare phenomenon in Germany; they can occur anywhere and are seasonally concentrated in the summer months. About 50 % of all heavy rainfall events between 2011 and 2021 can be spatially overlaid with the German rail network, and almost the entire rail network has been affected by a heavy rainfall event at least once during this 11-year period. Heavy rainfall events and associated natural hazards can therefore potentially affect the entire German rail network. However, vulnerability varies greatly from region to region and is determined, for example, by the route of the line in relation to the topography (Braud et al., 2020). Routes that follow valley courses or cross low mountain ranges are particularly susceptible to associated processes such as gravitational mass movements and local flooding. In order to make rail transport more resilient to heavy rainfall, it is important to gain a more detailed knowledge about cause-effect relationships between heavy rainfall events and the disruptions they trigger.

Often it is not the heavy rainfall event itself that cause damage to transport infrastructure, but processes that are triggered by them. Connections between heavy rainfall events as a triggering factor for further processes such as flooding (Bernet et al., 2019; Wake, 2013) and various types of gravitational mass movements (Araújo et al., 2022; Huggel et al., 2012; Kirschbaum et al., 2022; Tichavský et al., 2019) have already been established in several studies. Similarly, the regression models in our study show that when all meteorological variables are at their means, heavy rainfall events can in the two days following the event significantly increase the odds of occurrence of flood by a factor of 22.7 and tree fall events by a factor of 3.62 (see Table 3). The odds ratios of flood events decreases the more time passed after the heavy rainfall event, while the odds ratios of tree fall events peaks the day after a heavy rainfall event (Table 4). The increased odds of gravitational mass movement events is only statistically significant the day of and the day after a heavy rainfall event, but is also strongly correlated to precipitation and accumulated precipitation (Tables 2 and 4). It is therefore important not to consider the occurrence of different natural hazards individually, but to establish connections between the processes, for example by using climate impact chains (e.g. UBA, 2021) or a compound-hazard approach (e.g. Zscheischler et al., 2020).

About a quarter of all flood events could be coupled with a heavy rainfall event, and for gravitational mass movements it was as much as 17 % (Figure 1). The proportion of tree fall events connected to heavy rainfall events is very low, which could be due to the fact that storms and strong winds are considered the main trigger for this type of event (e.g. Bíl et al., 2017; Gardiner et al., 2010). A large proportion of the tree fall disturbances recorded in the DB damage database have been caused by a few large autumn and winter storms, such as Friederike in January 2018 (286 reports) or Sabine in February 2020 (513 reports), which were characterized by prolonged precipitation rather than heavy rainfall events. The influence of heavy rainfall on increasing the risk of tree fall has hardly been studied so far. Morimoto et al. (2021) found that heavy rainfall connected to typhoons increases the probability of disturbances in forest stands. Even if a spatial and temporal overlap of a heavy rainfall with an event from the damage database could be determined, it must be emphasized once again at this point that the heavy rainfall event can only be considered as a possible cause for the event and the actual causal trigger cannot be derived from the DB damage database. With our study, the aim is not to develop a predictive model of the natural hazards, but instead to provide empirical evidence of the potential relationship between heavy rainfall and the three natural hazard processes. We also demonstrated how damage data from infrastructure operators can be merged with climate data from weather services to establish a potential relationship. This step represents an important contribution in terms of proactive natural hazard management to identify the route sections that are particularly affected by certain climatic parameters and associated processes. Furthermore, this information can be used to prioritize adaptation needs.

The parameters Heavy Rainfall Index (SRI) and Antecedent Precipitation Index (V3) are the properties of the heavy rainfall events that most strongly influence the occurrence of all three natural hazard processes considered (Table 4, Figure 4). Thus, it is a combination of the pre-moisture conditions of the soil due to previous rainfall events and the occurrence of a heavy rainfall event, which most clearly promotes the occurrence of the processes. This is in concordance with, for example, findings from Rupp (2022), who analyzed the triggering factors for landslides with seasonal resolution. Antecedent precipitation is of great importance for the occurrence of landslides all year round, but especially in winter. Locosselli et al. (2021) found a similar seasonal variability for the climate drivers for tree falls among urban trees in Brazil. During the wet season, temperature has a direct influence on tree fall, while precipitation and wind gusts can have lagged effects.

No information on the magnitude of the hazard events can be obtained from the damage database. The duration of the disturbance, which is given for flood and tree fall events only, shows that for floods 33 % of the events have a disturbance duration of more than one day, for tree falls only 2 % (Fabella and Szymczak, 2021). From the rather short disruption durations, it can be deduced that most of the events must be smaller, as it is not possible to resume operations after a short time in the case of a larger event. In the case of smaller events, the local climate conditions, as represented for example by SRI and V3, are most important. Hence, no significant correlations could be observed with the larger-scale parameters such as mean precipitation, mean topographic position index and mean daily soil moisture. The role of the parameter degree of soil sealing (VSGL) on tree falls could be explained by the fact that areas with a high degree of sealing tend to have fewer trees along the

track that can potentially cause disturbances, while more rural and less sealed areas have more trees and therefore also an increased risk of tree fall events.

## 4.2 Data availability and quality

While the data quality of the CatRaRE catalogue is very high, it is difficult to validate the quality and completeness of the DB damage database. Therefore, it must be taken into account that the relatively low numbers of damage reports that could be
linked to a heavy rainfall event are only minimum values due to the weaknesses of the data collection process. While the DWD is responsible for meeting the meteorological needs of all economic and social sectors in Germany, the DB damage database is an internal product. The main task of a railroad operator is to ensure safe railroad operations. The focus is not on the detailed recording of the damage event with exact process allocation, cause, etc., but rather on enabling a quick repair and ensuring the resumption of railroad operations. However, disruptions caused by natural hazards account for a substantial proportion of
disruption events overall. In 2018, for example, weather-related disruptions were the second most frequent cause of cancellations according to DB data (Deutscher Bundestag, 2019). As climate change advances, it can be assumed that the number and extent of disruptive events is more likely to increase rather than decrease in the future, unless targeted countermeasures are taken. It is therefore essential to adapt rail transport and rail infrastructure to climate change. However, this requires reliable data on past damage events in order to guarantee a statistically robust consequence-based risk assessment
and the targeted development of measures for action in the future. We therefore recommend improving the documentation requirements for the various modes of transport in order to create a reliable damage database in the long-term. This should also include a subdivision of natural hazard events according to the underlying processes. For instance, river floods are typically caused by (longer) precipitation runoff in larger areas of the river watershed, while local flash floods are caused by the immediate runoff of concentrated, intense heavy rainfall events (Penna et al., 2013). Gravitational mass movements should
be classified according to their volume and type of transported materials, transportation processes and triggers, as e.g. heavy rainfall events typically trigger shallow landslides, while accumulated rainfall contributes more to deeper landslides (Zêzere et al., 2015).

## 4.3 Future development of heavy rainfall events and associated hazards

In Western and Central Europe, extreme rainfall has already increased in frequency and will, with high confidence, continue
to increase further with climate change (Seneviratne et al., 2021). However, modelling current and future trends in heavy rainfall events on a regional scale is a challenging task. Rybka et al. (2022) used a convection-permitting regional climate model to estimate return levels dependent on the rainfall duration and return period for Germany. They found a 30 % mean increase in intensity for daily rainfall extremes for the end of the 21$^{st}$ century assuming a high-end emission scenario, but the model shows no further increase in intensity for sub-daily heavy rainfall estimates. Although the exact rate is a subject of
debate, it can be assumed that with rising temperatures more water vapor can potentially be retained in the atmosphere, thus increasing the potential for the occurrence of heavy rainfall events (Lengfeld et al., 2021; Zeder and Fischer, 2020). Several

studies using observational data (e.g. Westra et al., 2013) or modeling experiments (e.g. O'Gorman, 2015) tested successfully the hypothesis that the intensity of daily extreme rainfall follows roughly the Clausis-Clapeyron relationship, e.g. an increase of roughly 7 % per °C ambient temperature (Allen and Ingram, 2002; Trenberth, 1999). An increase in daily (e.g. Westra et al., 2014; Fischer and Knutti, 2015) and sub-daily precipitation (e.g. Lenderink and Meijgaard, 2008; Guerreiro et al., 2018) extremes is already observed in several studies over many regions. Especially in the summer months, with a combination of long dry periods interrupted by single heavy precipitation periods, it can be assumed that these heavy rainfall events can lead to an increase of associated processes, e.g. landslides (Tichavský et al., 2019).

The timespan of the DB damage database it too short to analyze trends in the occurrence of the three types of natural hazards. Access to high quality data on past natural hazard-related disruptions in the transport sector is a major limitation and one of the reasons while there are only few scientific studies available on this issue (e.g. Braud et al., 2020; Donnini et al., 2017; Fabella and Szymczak, 2021; Gardiner et al. (in review)). However, quantifying the impact of natural hazards on the transport sector is of great importance, especially with regard to climate change. A global study by Koks et al. (2019) shows that already today about 27 % of all road and rail assets are exposed to at least one natural hazard. Climate change has a significant impact on forest stability (Seidl et al., 2017), and the frequency and magnitude of several natural hazards are likely to increase with ongoing climate change, as shown for gravitational mass movements (e.g. Chiang and Chang, 2011; Gariano and Guzzetti, 2016) or flash floods (e.g. Kundzewicz et al., 2013). It is therefore very likely that disturbances along transport routes due to natural hazards will occur more frequently in the future.

**5 Conclusions**

Due to the heavy rainfall event in July 2021 and the resulting flash floods and damage, awareness of vulnerability to this natural hazard has increased significantly and, among other things, a large number of research activities has been initiated. As the rail infrastructure was particularly hard hit, we contribute to raising awareness in the rail sector and in the transport sector in general with our study. We were able to show that heavy rainfall events have a significant influence on the occurrence of associated natural hazards. Furthermore, we demonstrate an approach to link climate data with damage data of a transport mode in order to establish a correlational interdependence. This can also be applied to other climate impacts and other modes of transport and represents an important component in the context of proactive natural hazard management.

**Data availability**

The CatRaRE data used for this study are available at https://www.dwd.de/DE/leistungen/catrare/catrare.html.

**Author contributions**

Conceptualisation: SS, FB, VF and KF; methodology: SS, FB, VF and KF; software: FB, VF and KF; data analysis and interpretation: SS, FB, VF and KF; writing – original draft preparation: SS; writing – review and editing: SS, FB, VF and KF; visualisation: FB, VF and KF. All authors have read and agreed to the published version of the manuscript.

**Competing interests**

The contact author has declared that none of the authors has any competing interests.

**Acknowledgments**

The authors thanks DB Netz AG for providing data from the damage data base.

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

**Appendix A**

In this Appendix we show the full results of the regression models including all explanatory variables, seasonal and year dummies and interaction terms. This section also addresses the question of overfitting in the models presented in the main text by presenting simpler models and assessing the robustness of the results.

Tables A1, A2 and A3 present the estimation results for the model in Table 2, for the natural hazards tree fall, flood and gravitational mass movement, respectively. Column (1) – (5) of each table are results of a random-effects logistic regression, beginning with a simple regression in column (1) and successively adding control variables and interaction terms in columns (2) – (4) until the final model is reached in (5), of which estimates are presented in the main text (Table 2). Columns (1) – (5) provides some insight into how sensitive the results are to changes in the selection of control variables. The simple logistic regression in column (1) would be less prone to overfitting due to having only one coefficient to estimate, but would be very prone to omitted variable bias. The omitted variable bias is evident from the fact that, in all three tables, the magnitude of the coefficient estimate for heavy rainfall dramatically changes once the control variables precipitation, accumulated precipitation and daily soil moisture are added to the model (column (2)), as well as when interaction terms are added to the model (column (4)).

Column (6) presents the results of a pooled logistic regression model, which is a regression of panel data where the time dimension is not considered, i.e. the data is regarded as cross-sectional. Column (6) addresses the concern that the panel data analysis, due to its two-dimensional nature – with the time dimension and the individual dimension (here route segments) – has many fixed-effects parameters to estimate, which may lead to overfitting. For all models that include interaction terms, the odds ratios of heavy rain at the mean values of the meteorological variables are reported at the bottom of the table.

In Table A1, column (1) indicates that the odds of a tree fall occurring is four times higher when there was heavy rainfall in the last three days versus no heavy rain. This effect, however, becomes less than one upon addition of the control variables precipitation, accumulated precipitation and daily soil moisture, as seen in columns (2) and (3). This means that keeping meteorological variables constant, the odds of a tree fall event is lower during heavy rain versus when no heavy rain occurs. In columns (4) and (5), the interactions between heavy rain and the control variables are included, and this causes an even bigger jump in the coefficient estimate of heavy rain, indicating that heavy rain indeed has an effect on the incidence of tree fall, and this effect varies depending on the meteorological situation. The pooled logit estimates in column (6) do not differ significantly from the results of the full model in column (5), suggesting that the unobserved individual heterogeneity of the route segments does not play a crucial role in determining the relationship between heavy rainfall and tree fall incidence. The odds ratios computed based on the means of the meteorological variables in columns (4) to (6) are of similar magnitude to the coefficient of the simple logistic regression in column (1). If a simpler model were selected to potentially avoid overfitting,

say the model in column (1), then the results would not differ by very much. However, model (5) is preferred because it overcomes the omitted variable bias and provides a more nuanced picture of the complex relationship between heavy rainfall and the incidence of tree fall.

For flood hazards (Table A2), the odd ratio of heavy rain is close to or greater than 10 for all models, indicating that the result is robust to the choice of controls. When comparing the odds ratio in model (1) to the odds ratio at the means in models (4) – (6), one can observe that, although similar in magnitude, the value in model (1) is still somewhat larger. This could suggest that for floods in particular, the omitted variable bias is quite substantial, and that including meteorological controls is crucial. The smaller coefficient of the pooled logit in column (6) compared to the full model in column (5) suggests that for flood

events, unobserved individual heterogeneity in the route segments (for example inclination) may influence the effect of heavy rain on flood incidence.

For gravitational mass movements (Table A3), the magnitude of the odds ratio in the simple regression in column (1) are of the same magnitude as the odds ratios at the mean (at the bottom of columns (2) to (6)) once interactions are added. However,

with the addition of interaction terms, the statistical significance of heavy rain disappears. One could perceive this as potentially indicating overfitting, but the model in column (4) with interactions is a simpler model with fewer parameters than the model in column (3) without interactions. Model complexity alone did not cause the coefficient of heavy rain to lose its significance, which could suggest that it is rather omitted variable bias that is at play here. Nevertheless, one could argue that a simpler model, say in column (1) might still be preferable, particularly for gravitational mass movements, which is at most risk of

overfitting due to the low number of events in the data. This will not change the interpretation by very much, since the odds ratio in model (1) and the odds ratios at the means in model (5) are similar in magnitude. However, from these results alone, one cannot conclude a statistically robust effect of heavy rainfall on gravitational mass movements. More data is required to make solid conclusions.

In all three tables A1, A2 and A3, the model with the best fit, i.e. the highest log likelihood and the lowest AIC, is the model in column (5), which are the results presented in Table 2 of the main text.

Tables A4, A5 and A6 provide estimation results for the model in Table 4 of the main text, where a dummy variable for the day of the heavy rain, one day after heavy rain and two days after heavy rain are the variables of interest. In these tables,

columns (1) – (3) have estimates of a random-effects logit with successively additional control variables, while column (4) has estimates of a pooled logit regression of the full model. The results in column (3) of Tables A3, A4, and A5 are the same as those presented in Table 4. Similar to the previous tables, a considerable jump in the odds ratios can be seen between columns (1) and (2), potentially due to omitted variable bias. However, for floods and gravitational mass movements, the direction of relative effects across time remains the same for all models despite the differences in magnitudes. For flood events (Table A5),

the odds of having a flood is highest on the same day as the heavy rainfall, and decreases the further away in time the rainfall occurred. For gravitational mass movements (Table A6), the odds are highest one day after heavy rainfall occurred. In the case where overfitting is a potential issue, such as for gravitational mass movements, a simpler model as in model (2) might be preferred to the one presented in the main text (Table 4). This would not change the result that the highest odds of a mass movement event comes one day after heavy rain.


    For tree fall (Table A4), both the magnitude of the effect and the direction across time changes once control variables are added. Without the control variables (column (1)), the odds of tree fall are highest on the same day as the occurrence of heavy rainfall, and decreases in succeeding days. However, once the meteorological variables are added, the odds of a tree fall event on the same day as the heavy rain are in fact lower than the odds of when there is no heavy rain. This can be seen in columns

(2) – (4) of Table A4, where the odds ratios on the day of heavy rain are less than one. One day after heavy rain, on the other hand, the odds of a tree fall event are around twice as high as on days without heavy rain, and the effect is highest on this day.

    In all three tables A4, A5 and A6, the goodness-of-fit parameters performed best in the full model in column (3), whose results are presented in the main text.


    In summary, there is a trade-off between omitted variable bias and overfitting in this empirical analysis, and the results presented in the main text are the ones deemed most appropriate for the investigation of the relationship between heavy rainfall and the incidence of natural hazards along railway lines. In the cases most at risk of overfitting due to the low number of events, particularly for gravitational mass movements, the interpretation of the results would not change if a simpler model

were to be selected.

**Table A1. Tree fall: Random-effects and pooled logistic regressions with successive inclusion of control variables**

| | Dependent Variable: Tree Fall | | | | | |
|---|---|---|---|---|---|---|
| | Random-Effects Logit | | | | | Pooled Logit |
| | (1) | (2) | (3) | (4) | (5) | (6) |
| Heavy rain, last 3 days=1 | 4.313*** | 0.651*** | 0.620*** | 31.53*** | 39.85*** | 29.22*** |
| | (0.282) | (0.0797) | (0.0739) | (23.67) | (29.91) | (21.51) |
| Precipitation at route segment [mm] | | 1.049*** | 1.047*** | 1.071*** | 1.069*** | 1.067*** |
| | | (0.00138) | (0.00135) | (0.00116) | (0.00117) | (0.00109) |
| Accumulated precipitation at route segment, 30 days [mm] | | 1.006*** | 1.004*** | 1.005*** | 1.003*** | 1.003*** |
| | | (0.000266) | (0.000290) | (0.000269) | (0.000293) | (0.000305) |
| Daily soil moisture at route segment [% nFK] | | 0.926*** | 0.927*** | 0.930*** | 0.931*** | 0.936*** |
| | | (0.00302) | (0.00316) | (0.00303) | (0.00316) | (0.00314) |
| Daily soil moisture at route segment [% nFK], squared | | 1.001*** | 1.001*** | 1.001*** | 1.001*** | 1.001*** |
| | | (0.0000219) | (0.0000228) | (0.0000218) | (0.0000226) | (0.0000225) |
| Spring | | | 1.197*** | | 1.189*** | 1.184*** |
| | | | (0.0331) | | (0.0327) | (0.0333) |
| Autumn | | | 2.056*** | | 1.971*** | 2.008*** |
| | | | (0.0838) | | (0.0796) | (0.0838) |
| Winter | | | 1.231*** | | 1.209*** | 1.228*** |
| | | | (0.0396) | | (0.0388) | (0.0409) |
| Year=2018 | | | 1.052 | | 1.046 | 1.033 |
| | | | (0.0294) | | (0.0291) | (0.0287) |
| Year=2019 | | | 0.942* | | 0.940* | 0.926** |
| | | | (0.0274) | | (0.0273) | (0.0267) |
| Year=2020 | | | 0.948 | | 0.945* | 0.936* |
| | | | (0.0266) | | (0.0263) | (0.0261) |
| Heavy rain, last 3 days=1 x Precipitation at route segment [mm] | | | | 0.940*** | 0.942*** | 0.943*** |
| | | | | (0.00257) | (0.00257) | (0.00250) |
| Heavy rain, last 3 days=1 x Accumulated precipitation at route segment, 30 days [mm] | | | | 0.997* | 0.997* | 0.997* |
| | | | | (0.00130) | (0.00137) | (0.00135) |
| Heavy rain, last 3 days=1 x Daily soil moisture at route segment [% nFK] | | | | 0.967 | 0.959* | 0.969 |
| | | | | (0.0195) | (0.0196) | (0.0192) |
| Heavy rain, last 3 days=1 x Daily soil moisture at route segment [% nFK], squared | | | | 1.000 | 1.000 | 1.000 |
| | | | | (0.000126) | (0.000128) | (0.000124) |
| Observations | 14141019 | 14141019 | 14141019 | 14141019 | 14141019 | 14141019 |
| Odds ratio of heavy rain at mean values of the meteorological variables | | | | 4.008 | 3.616 | 3.730 |
| Log likelihood | -90610.2 | -88449.3 | -88238.6 | -88044.4 | -87853.1 | -92607.7 |
| rho | 0.385 | 0.377 | 0.376 | 0.376 | 0.375 | |
| AIC | 181226.3 | 176912.7 | 176503.2 | 176110.9 | 175740.3 | 185247.3 |

Exponentiated coefficients (odds ratios); Standard errors in parentheses; $^{*}\, p < 0.05$, $^{**}\, p < 0.01$, $^{***}\, p < 0.001$

 **Table A2. Flood: Random-effects and pooled logistic regressions with successive inclusion of control variables**

| | Dependent Variable: Floods | | | | | |
| --- | --- | --- | --- | --- | --- | --- |
| | Random-Effects Logit | | | | | Pooled Logit |
| | (1) | (2) | (3) | (4) | (5) | (6) |
| Heavy rain, last 3 days=1 | 76.61*** | 11.31*** | 9.245*** | 29.19** | 34.29** | 19.52* |
| | (5.545) | (1.302) | (1.105) | (36.32) | (41.71) | (24.64) |
| Precipitation at route segment [mm] | | 1.027*** | 1.026*** | 1.082*** | 1.079*** | 1.078*** |
| | | (0.00152) | (0.00157) | (0.00357) | (0.00360) | (0.00347) |
| Accumulated precipitation at route segment, 30 days [mm] | | 1.014*** | 1.011*** | 1.013*** | 1.010*** | 1.008*** |
| | | (0.000628) | (0.000787) | (0.000703) | (0.000843) | (0.000782) |
| Daily soil moisture at route segment [% nFK] | | 0.950*** | 0.953*** | 0.948*** | 0.944*** | 0.952*** |
| | | (0.0115) | (0.0120) | (0.0140) | (0.0138) | (0.0137) |
| Daily soil moisture at route segment [% nFK] , squared | | 1.000*** | 1.000*** | 1.000*** | 1.000*** | 1.000*** |
| | | (0.0000768) | (0.0000796) | (0.0000936) | (0.0000939) | (0.0000918) |
| Spring | | | 1.867*** | | 1.881*** | 1.706*** |
| | | | (0.196) | | (0.204) | (0.183) |
| Autumn | | | 3.098*** | | 2.812*** | 2.575*** |
| | | | (0.409) | | (0.372) | (0.322) |
| Winter | | | 1.156 | | 1.155 | 1.022 |
| | | | (0.152) | | (0.151) | (0.133) |
| Year=2012 | | | 1.029 | | 1.027 | 0.984 |
| | | | (0.203) | | (0.201) | (0.190) |
| Year=2013 | | | 1.151 | | 1.208 | 1.290 |
| | | | (0.189) | | (0.200) | (0.213) |
| Year=2014 | | | 1.327 | | 1.289 | 1.288 |
| | | | (0.242) | | (0.234) | (0.230) |
| Year=2015 | | | 0.610* | | 0.610* | 0.591* |
| | | | (0.144) | | (0.143) | (0.138) |
| Year=2016 | | | 1.321 | | 1.321 | 1.314 |
| | | | (0.233) | | (0.231) | (0.230) |
| Year=2017 | | | 1.583** | | 1.569** | 1.617** |
| | | | (0.265) | | (0.262) | (0.269) |
| Year=2018 | | | 3.169*** | | 3.111*** | 2.900*** |
| | | | (0.554) | | (0.538) | (0.490) |
| Year=2019 | | | 2.487*** | | 2.479*** | 2.303*** |
| | | | (0.434) | | (0.431) | (0.389) |
| Year=2020 | | | 2.170*** | | 2.208*** | 2.050*** |
| | | | (0.365) | | (0.369) | (0.335) |
| Year=2021 | | | 2.518*** | | 2.719*** | 2.795*** |
| | | | (0.391) | | (0.414) | (0.422) |
| Heavy rain, last 3 days=1 x Precipitation at route segment [mm] | | | | 0.942*** | 0.943*** | 0.944*** |
| | | | | (0.00343) | (0.00346) | (0.00332) |
| Heavy rain, last 3 days=1 x Accumulated precipitation at route segment, 30 days [mm] | | | | 0.998 | 0.999 | 0.999 |
| | | | | (0.00111) | (0.00125) | (0.00120) |
| Heavy rain, last 3 days=1 x Daily soil moisture at route segment [% nFK] | | | | 1.006 | 1.002 | 1.017 |
| | | | | (0.0300) | (0.0298) | (0.0310) |
| Heavy rain, last 3 days=1 x Daily soil moisture at route segment [% nFK], squared | | | | 1.000 | 1.000 | 1.000 |
| | | | | (0.000176) | (0.000179) | (0.000182) |
| Observations | 38590173 | 38590173 | 38590173 | 38590173 | 38590173 | 38590173 |
| Odds ratio of heavy rain at mean values of the meteorological variables | | | | 27.47 | 22.70 | 24.04 |
| Log likelihood | -11514.0 | -10923.3 | -10780.6 | -10787.1 | -10645.3 | -10907.5 |
| rho | 0.373 | 0.435 | 0.437 | 0.428 | 0.430 | |
| AIC | 23034.0 | 21860.7 | 21601.3 | 21596.1 | 21338.6 | 21861.1 |

Exponentiated coefficients (odds ratios); Standard errors in parentheses; $^{*} p < 0.05$, $^{**} p < 0.01$, $^{***} p < 0.001$

**Table A3. Gravitational Mass Movement: Random-effects and pooled logistic regressions with successive inclusion of control variables**

| | Dependent Variable: Gravitational Mass Movements | | | | | |
| | Random-Effects Logit | | | | | Pooled Logit |
| | (1) | (2) | (3) | (4) | (5) | (6) |
|---|---|---|---|---|---|---|
| Heavy rain, last 3 days=1 | 28.53*** | 5.298*** | 4.795*** | 9.206 | 3.812 | 3.188 |
| | (4.646) | (1.281) | (1.184) | (26.63) | (11.26) | (9.426) |
| Precipitation at route segment [mm] | | 1.018*** | 1.018*** | 1.054*** | 1.052*** | 1.050*** |
| | | (0.00331) | (0.00343) | (0.00680) | (0.00691) | (0.00686) |
| Accumulated precipitation at route segment, 30 days [mm] | | 1.015*** | 1.014*** | 1.015*** | 1.014*** | 1.013*** |
| | | (0.00105) | (0.00126) | (0.00113) | (0.00128) | (0.00124) |
| Daily soil moisture at route segment [% nFK] | | 0.985 | 0.980 | 0.972 | 0.957 | 0.960 |
| | | (0.0216) | (0.0227) | (0.0231) | (0.0233) | (0.0245) |
| Daily soil moisture at route segment [% nFK], squared | | 1.000 | 1.000 | 1.000 | 1.000* | 1.000* |
| | | (0.000137) | (0.000145) | (0.000149) | (0.000153) | (0.000161) |
| Spring | | | 1.845*** | | 1.922*** | 1.799*** |
| | | | (0.300) | | (0.308) | (0.287) |
| Autumn | | | 1.648* | | 1.630* | 1.488 |
| | | | (0.360) | | (0.349) | (0.318) |
| Winter | | | 0.819 | | 0.844 | 0.778 |
| | | | (0.173) | | (0.179) | (0.163) |
| Year=2014 | | | 0.796 | | 0.747 | 0.747 |
| | | | (0.169) | | (0.157) | (0.158) |
| Year=2015 | | | 1.054 | | 1.007 | 0.948 |
| | | | (0.221) | | (0.208) | (0.195) |
| Year=2016 | | | 1.059 | | 0.999 | 0.990 |
| | | | (0.211) | | (0.197) | (0.194) |
| Year=2017 | | | 0.434** | | 0.409*** | 0.398*** |
| | | | (0.111) | | (0.104) | (0.101) |
| Year=2018 | | | 0.822 | | 0.763 | 0.765 |
| | | | (0.193) | | (0.177) | (0.176) |
| Year=2019 | | | 0.893 | | 0.846 | 0.796 |
| | | | (0.201) | | (0.188) | (0.176) |
| Year=2020 | | | 0.751 | | 0.706 | 0.683 |
| | | | (0.171) | | (0.159) | (0.153) |
| Year=2021 | | | 0.986 | | 1.000 | 0.992 |
| | | | (0.196) | | (0.192) | (0.194) |
| Heavy rain, last 3 days=1 x Precipitation at route segment [mm] | | | | 0.955*** | 0.956*** | 0.959*** |
| | | | | (0.00781) | (0.00792) | (0.00798) |
| Heavy rain, last 3 days=1 x Accumulated precipitation at route segment, 30 days [mm] | | | | 0.994* | 0.996 | 0.995 |
| | | | | (0.00269) | (0.00280) | (0.00267) |
| Heavy rain, last 3 days=1 x Daily soil moisture at route segment [% nFK] | | | | 1.036 | 1.061 | 1.070 |
| | | | | (0.0707) | (0.0742) | (0.0749) |
| Heavy rain, last 3 days=1 x Daily soil moisture at route segment [% nFK], squared | | | | 1.000 | 1.000 | 0.999 |
| | | | | (0.000394) | (0.000405) | (0.000406) |
| Observations | 31795515 | 31795515 | 31795515 | 31795515 | 31795515 | 31795515 |
| Odds ratio of heavy rain at mean values of the meteorological variables | | | | 19.31 | 17.90 | 19.50 |
| Log likelihood | -4542.7 | -4379.4 | -4352.0 | -4351.3 | -4322.7 | -4462.2 |
| rho | 0.519 | 0.535 | 0.536 | 0.531 | 0.531 | |
| AIC | 9091.4 | 8772.9 | 8740.0 | 8724.7 | 8689.5 | 8966.5 |

Exponentiated coefficients (odds ratios); Standard errors in parentheses; $^{*} p < 0.05$, $^{**} p < 0.01$, $^{***} p < 0.001$

**Table A4. Tree fall: Random-effects and pooled logistic regressions different numbers of days after a heavy rainfall event.**

| | Dependent Variable: Tree Fall | | | |
| --- | --- | --- | --- | --- |
| | Random-Effects Logit | | | Pooled Logit |
| | (1) | (2) | (3) | (4) |
| Days from heavy rainfall event | | | | |
| - day of heavy rain | 5.784*** | 0.339*** | 0.333*** | 0.310*** |
| | (0.457) | (0.0547) | (0.0525) | (0.0511) |
| - 1 day after heavy rain | 4.223*** | 2.702*** | 2.411*** | 2.252*** |
| | (0.514) | (0.341) | (0.305) | (0.282) |
| - 2 days after heavy rain | 1.569* | 1.043 | 0.963 | 0.934 |
| | (0.315) | (0.212) | (0.195) | (0.188) |
| Precipitation at route segment [mm] | | 1.054*** | 1.053*** | 1.050*** |
| | | (0.00152) | (0.00150) | (0.00147) |
| Accumulated precipitation at route segment, 30 days [mm] | | 1.006*** | 1.004*** | 1.003*** |
| | | (0.000271) | (0.000294) | (0.000303) |
| Daily soil moisture at route segment [% nFK] | | 0.926*** | 0.928*** | 0.933*** |
| | | (0.00302) | (0.00316) | (0.00313) |
| Daily soil moisture at route segment [% nFK], squared | | 1.001*** | 1.001*** | 1.001*** |
| | | (0.0000218) | (0.0000228) | (0.0000226) |
| Spring | | | 1.194*** | 1.192*** |
| | | | (0.0330) | (0.0336) |
| Autumn | | | 2.035*** | 2.083*** |
| | | | (0.0829) | (0.0878) |
| Winter | | | 1.224*** | 1.246*** |
| | | | (0.0395) | (0.0417) |
| Year=2018 | | | 1.051 | 1.043 |
| | | | (0.0294) | (0.0292) |
| Year=2019 | | | 0.944* | 0.933* |
| | | | (0.0275) | (0.0270) |
| Year=2020 | | | 0.948 | 0.939* |
| | | | (0.0266) | (0.0264) |
| Observations | 14141019 | 14141019 | 14141019 | 14141019 |
| Log likelihood | -90585.0 | -88384.7 | -88178.9 | -92952.3 |
| Rho | 0.385 | 0.377 | 0.376 | |
| AIC | 181180.0 | 176787.4 | 176387.9 | 185932.5 |

Exponentiated coefficients (odds ratios); Standard errors in parentheses; $^{*}p < 0.05$, $^{**}p < 0.01$, $^{***}p < 0.001$


**Table A5. Flood: Random-effects and pooled logistic regressions different numbers of days after a heavy rainfall event.**

| | Dependent Variable: Floods | | | |
| --- | --- | --- | --- | --- |
| | Random-Effects Logit | | | Pooled Logit |
| | (1) | (2) | (3) | (4) |
| Days from heavy rainfall event | | | | |
| - day of heavy rain | 122.6*** | 13.55*** | 11.41*** | 11.85*** |
| | (10.03) | (2.159) | (1.869) | (1.851) |
| - 1 day after heavy rain | 46.04*** | 12.23*** | 9.529*** | 9.827*** |
| | (7.117) | (2.058) | (1.620) | (1.651) |
| - 2 days after heavy rain | 20.44*** | 5.821*** | 4.757*** | 4.897*** |
| | (4.618) | (1.328) | (1.082) | (1.108) |
| Precipitation at route segment [mm] | | 1.025*** | 1.023*** | 1.021*** |
| | | (0.00212) | (0.00214) | (0.00186) |
| Accumulated precipitation at route segment, 30 days [mm] | | 1.015*** | 1.011*** | 1.009*** |
| | | (0.000627) | (0.000794) | (0.000717) |
| Daily soil moisture at route segment [% nFK] | | 0.952*** | 0.955*** | 0.970* |
| | | (0.0115) | (0.0120) | (0.0120) |
| Daily soil moisture at route segment [% nFK], squared | | 1.000*** | 1.000*** | 1.000*** |
| | | (0.0000763) | (0.0000791) | (0.0000784) |
| Spring | | | 1.847*** | 1.654*** |
| | | | (0.195) | (0.174) |
| Autumn | | | 3.058*** | 2.827*** |
| | | | (0.404) | (0.354) |
| Winter | | | 1.150 | 1.019 |
| | | | (0.151) | (0.134) |
| Year=2012 | | | 1.034 | 0.989 |
| | | | (0.204) | (0.191) |
| Year=2013 | | | 1.151 | 1.184 |
| | | | (0.189) | (0.192) |
| Year=2014 | | | 1.332 | 1.326 |
| | | | (0.243) | (0.236) |
| Year=2015 | | | 0.615* | 0.594* |
| | | | (0.145) | (0.138) |
| Year=2016 | | | 1.333 | 1.310 |
| | | | (0.236) | (0.230) |
| Year=2017 | | | 1.589** | 1.639** |
| | | | (0.266) | (0.272) |
| Year=2018 | | | 3.199*** | 2.982*** |
| | | | (0.560) | (0.507) |
| Year=2019 | | | 2.496*** | 2.321*** |
| | | | (0.436) | (0.392) |
| Year=2020 | | | 2.185*** | 2.005*** |
| | | | (0.368) | (0.328) |
| Year=2021 | | | 2.563*** | 2.614*** |
| | | | (0.397) | (0.398) |
| Observations | 38590173 | 38590173 | 38590173 | 38590173 |
| Log likelihood | -11457.4 | -10916.5 | -10773.6 | -11041.5 |
| Rho | 0.374 | 0.435 | 0.437 | |
| AIC | 22924.8 | 21851.1 | 21591.2 | 22125.1 |

Exponentiated coefficients (odds ratios); Standard errors in parentheses; $^{*} p < 0.05$, $^{**} p < 0.01$, $^{***} p < 0.001$

**Table A5. Gravitational Mass Movement: Random-effects and pooled logistic regressions different numbers of days after a heavy rainfall event.**

| | Dependent Variable: Gravitational Mass Movements | | | |
|---|---|---|---|---|
| | Random-Effects Logit | | | Pooled Logit |
| | (1) | (2) | (3) | (4) |
| Days from heavy rainfall event | | | | |
| - day of heavy rain | 33.01*** | 3.973*** | 3.584*** | 3.414** |
| | (6.775) | (1.521) | (1.387) | (1.316) |
| - 1 day after heavy rain | 44.11*** | 11.71*** | 10.58*** | 10.16*** |
| | (10.44) | (3.067) | (2.842) | (2.760) |
| - 2 days after heavy rain | 4.398* | 1.205 | 1.120 | 1.101 |
| | (3.137) | (0.869) | (0.810) | (0.796) |
| Precipitation at route segment [mm] | | 1.022*** | 1.021*** | 1.021*** |
| | | (0.00448) | (0.00466) | (0.00432) |
| Accumulated precipitation at route segment, 30 days [mm] | | 1.015*** | 1.014*** | 1.013*** |
| | | (0.00106) | (0.00127) | (0.00124) |
| Daily soil moisture at route segment [% nFK] | | 0.987 | 0.982 | 0.989 |
| | | (0.0217) | (0.0228) | (0.0245) |
| Daily soil moisture at route segment [% nFK], squared | | 1.000 | 1.000 | 1.000 |
| | | (0.000137) | (0.000146) | (0.000155) |
| Spring | | | 1.829*** | 1.700*** |
| | | | (0.297) | (0.273) |
| Autumn | | | 1.622* | 1.502 |
| | | | (0.354) | (0.327) |
| Winter | | | 0.811 | 0.753 |
| | | | (0.172) | (0.158) |
| Year=2014 | | | 0.788 | 0.804 |
| | | | (0.167) | (0.172) |
| Year=2015 | | | 1.044 | 1.008 |
| | | | (0.218) | (0.211) |
| Year=2016 | | | 1.053 | 1.061 |
| | | | (0.209) | (0.209) |
| Year=2017 | | | 0.430*** | 0.427*** |
| | | | (0.110) | (0.108) |
| Year=2018 | | | 0.815 | 0.839 |
| | | | (0.191) | (0.195) |
| Year=2019 | | | 0.887 | 0.855 |
| | | | (0.199) | (0.190) |
| Year=2020 | | | 0.745 | 0.728 |
| | | | (0.169) | (0.164) |
| Year=2021 | | | 0.970 | 0.992 |
| | | | (0.193) | (0.200) |
| Observations | 31795515 | 31795515 | 31795515 | 31795515 |
| Log likelihood | -4533.6 | -4370.2 | -4343.0 | -4485.1 |
| Rho | 0.519 | 0.535 | 0.536 | |
| AIC | 9077.1 | 8758.5 | 8726.0 | 9008.1 |

Exponentiated coefficients (odds ratios); Standard errors in parentheses; $^*p < 0.05$, $^{**}p < 0.01$, $^{***}p < 0.001$