# Peer review of "Are heavy rainfall events a major trigger of associated natural hazards along the German rail network?"

_Natural Hazards and Earth System Sciences, 2023_

## Referee Comment (RC2)

**Manuscript Number:** nhess-2023-196

**Title:** Are heavy rainfall events a major trigger of associated natural hazards along the German rail network?

**Reviewer:** Ugur Ozturk

**Overview and general comments:**

The manuscript demonstrates the integration of damage data from infrastructure operators with climate data from weather services, aiming to discern potential relationships that could enhance proactive management of natural hazards. The authors' perspective, primarily through the lens of a railroad operator, brings a focused approach to understanding and mitigating disruptions to railroad operations caused by climate extremes. This perspective is particularly timely, given the anticipated increase in such disruptions under the impact of climate change, highlighting the urgent need for targeted countermeasures.

As I read the manuscript, I found the unique approach to examining rainfall, associated hazards, and their impact on the rail network from a rail network operator's standpoint to be both enlightening and compelling. The entire analysis bears the imprint of this distinctive viewpoint, offering insights that are both practical and relevant to the field. However, I feel that certain aspects of the study and the choices made therein would benefit from additional elucidation, and there may be room to broaden some analyses to further strengthen the findings and their implications.

My **major concerns** concentrate around the method choices forming the foundation of the current study. I highlight the line number of a piece from the manuscript in quotation marks which is followed by my comments after the sign of -->.

**Line 79:** "spatially intersected with the German rail network" --> Is this intersection achieved considering purely spatial overlap, or are rainfall runoff conditions taken into account as well? For instance, rainfall upstream could potentially impact tracks downstream, even in the absence of local precipitation.

**Line 97:** "one heavy rainfall event" --> Could the authors clarify if they are referring to hydrogeomorphological events, including mass-wasting process? I suspect that the tree falls might relate more to wind than rainfall. If the tree fall process is indeed related to wind, it might be beneficial to consider a term that encompasses all three phenomena. Perhaps including wind events as a factor, or alternatively, reconsidering the inclusion of tree fall cases, might provide a more concise, easy-to-explain analysis.

**Line 158:** "e.g. shown for shallow landslides" --> While the observation may hold true for shallow landslides, it's important to note that gravitational mass movements also encompass deep-seated landslides, where the lag time could extend significantly, potentially reaching years. Even excluding these exceptional cases, a lag time of 10-15 days appears realistic, as evidenced by Dille et al. (2022; https://doi.org/10.1038/s41561-022-01073-3). Should the focus be on shallow landslides, it would be helpful if this distinction is made clear. The choice of a 2-day lag for considering landslides raises some concerns for me, and I kindly suggest revisiting this aspect for a more nuanced discussion. The term "gravitational mass movements" might cover more processes than the authors intended.

**Lines 200–202:** "The segment is considered to have been affected by a heavy rainfall event on a given day if a heavy rainfall event from the CatRaRE database has occurred on that day up to a maximum of two days previously." --> As I mentioned in my previous comment, I'm

concerned that the proposed time-lag window might not adequately capture the lag time associated with landslides. Later on, in the results (below comment) authors highlight that only a small fraction of mass wasting events were linked to certain rain events. Hence, as previously mentioned, extending this window could offer a more accurate representation of the impact of heavy rainfall on landslide occurrences.

**Line 275:** "a total of 59 events (14 %)" --> I wonder if the correlation might become more pronounced if the lag time were extended to 15 days or more. This adjustment could potentially offer a more comprehensive analysis of the impact of heavy rainfall on these events.

**Lines 278-279:** "Of the 14461 tree fall events, a total of 312 (2 %) events can be spatially and temporally linked to a heavy rainfall event." --> This observation might suggest a less direct connection between rainfall and tree falls, potentially implicating other factors such as wind (as discussed by Gardiner et al.; http://dx.doi.org/10.2139/ssrn.4576016) or flooding (Lucia et al., 2018; https://doi.org/10.1016/j.scitotenv.2018.05.186). Further exploration of these factors could enrich the study.

My **minor concerns** primarily revolve around the use of terminology and the occasional absence of detailed explanatory statements that could further enhance the manuscript's readability and comprehension. Clarifying these aspects could improve the overall understandability for the readers. I list the minor comments in the attached file to keep my online comments concise.

**Line 6:** "associated natural hazards" --> Could you please specify which natural hazards are being referred to here?

**Line 8:** "random-effects logistic models" --> At this juncture, I'm finding it challenging to grasp the specifics of this model. Could you possibly elaborate further for clarity?

**Line 8:** "DB Netz AG" --> I'm concerned that this acronym might not be readily understood by a significant portion of the NHESS readership outside of Germany. Could a brief explanation be provided for broader accessibility?

**Line 8:** "CatRaRe" --> Similarly, this acronym might not resonate with the wider NHESS audience. A definition could greatly aid in understanding.

**Lines 10-11:** "Twenty-three percent of the flood events, 14% of the gravitational mass movements and 2% of the tree fall events" --> I'm having a bit of difficulty following these percentages. To clarify, does this imply that 77% of the flood events are not attributed to heavy rainfall?

**Lines 12-13:** "a heavy rainfall event significantly increases the probability of occurrence of a flood by a factor of 34.29." --> Am I correct in understanding that, according to the authors, there are floods that occur independently of heavy rainfall events?

**Line 12:** "Tree fall" --> I'm struggling to conceptualize how heavy rainfall directly leads to tree fall. Could you provide further insight into this connection?

**Lines 13-14:** "the 21-days antecedent precipitation index" --> I'm not entirely versed in the conventional determination of the 21-day threshold. If it's based on specific constraints, might I suggest clarifying or possibly reevaluating its presentation in the text?

**Line 15:** "with no significant increase for gravitational mass movements" --> This finding seems to be at odds with existing literature. Could the authors confidently assert the robustness of their data in this regard? Further emphasis on this point might be warranted,

especially given the indicated positive correlation between increasing rainfall and landslide occurrences in Figure 4.

**Line 15:** "21 day threshold" --> Once again, the 21-day threshold's basis is not clear to me. If it's a discretionary choice, clarifying its rationale or considering its removal for a more straightforward explanation might be beneficial.

**Lines 15-16:** "The results underline the importance of gaining more precise knowledge about the impact of climate triggers on natural hazard-related disturbances" --> The connection to this conclusion in the abstract seems somewhat tenuous. A more detailed exposition in the text might help in directly leading the reader to this message. Alternatively, authors could consider revising the final message that summarizes their results.

**Lines 19-26:** "" --> This is a really nice start to the manuscript. It really caught my attention.

**Line 28:** "significant" --> I would advise against using 'significant' unless it is in a statistical context. Perhaps 'considerably' could serve as a suitable alternative.

**Line 39:** "Within the framework of proactive natural hazard management" --> Is this framework well-documented in the literature, or is it an innovative concept proposed by the authors? A reference or a detailed explanation would be valuable for readers.

**Line 98:** "Starkregenindex SRI" --> Could the authors kindly provide a citation for this index? If it is a novel introduction, a detailed explanation within the manuscript would enhance understanding.

**Line 100:** "Lower Saxony" --> Given the frequent references to federal states, it may be helpful to include a reference map for those less familiar with Germany's geographical and political landscape.

**Line 111:** "with event-specific search terms" --> Could you specify which search terms were used?

**Line 128:** "the period 1 January 2017-16 December 2020" --> I'm curious as to why data from earlier years and for 2021 were not included. This query also pertains to mass-wasting events. If they were absent in the data, do the authors know the reason behind.

**Line 132:** "Explanatory control variables" --> It would be immensely helpful to visualize these data. Is it possible to include them in one or more figures within the manuscript?

**Lines 153-154:** "Thus, there are event locations where more than 50 heavy rainfall events from 2011 to 2021 can be found." --> Could this indicate that a single event, such as gravitational mass movements, might be attributed to multiple heavy rain events? This clarification would greatly aid in understanding the methodology applied.

**Line 161:** "for deep landslides" --> It may be more accurate to refer to these as "deep-seated landslides" to ensure clarity and precision in terminology.

**Line 166:** "the data set" --> It appears the term "dataset" is used inconsistently. Adopting a uniform usage could enhance the manuscript's readability.

**Line 270:** "events" --> Could you specify which events are being referred to here? Are these meant to be natural hazards?

**Lines 322-323:** "After two days, the probability of a gravitational mass movement is no longer different from a situation with no heavy rainfall." -->

**Lines 325-326:** "In contrast, one day after a heavy rainfall event, a tree fall event is 2.4 times more likely to occur than in days with no heavy rainfall." --> The linkage of tree falls to heavy rainfall events, as highlighted here, is quite intriguing, especially considering that 98% of tree

fall events were not directly linked to rainfall in the previous sections. Further clarification on this discrepancy would be beneficial.

**Lines 352–353:** "In the case of b) and e), the distance becomes greater at higher values, i.e., the higher the amount of accumulated precipitation, the more a heavy rainfall event increases the probability of occurrence of a flood or gravitational mass movement." --> It might be worthwhile for the authors to explore the findings of Saito et al. (2014; https://doi.org/10.1130/G35680.1), which suggest that increasing rainfall totals enhance landslide activity up to a certain threshold beyond which the effect does not apply. This could provide valuable insights into the study's analysis.

**Line 398:** "susceptible" --> The use of "susceptible" might be slightly misleading, potentially being confused with the term 'landslide susceptibility.' Perhaps "prone" could be a more precise alternative, avoiding any ambiguity.

**Lines 414-415:** "for gravitational mass movements it was as much as 17 % (Figure 1)" --> The low percentage of gravitational mass movements attributed to heavy rainfall seems somewhat unexpected, given that rainfall or related floods are the main triggers for landslides in Germany. This observation suggests the 2-day threshold might be too restrictive. Extending this threshold could potentially reveal a more nuanced impact of rainfall on landslide occurrence. It's commonly suggested that lag times of 10 to 15 days are realistic, with some extreme cases even longer, which could be an important consideration for the study.

**Figures:**

**Figure 1** --> The resemblance between subplot d and f is quite notable. This observation raises an interesting question: might the authors consider the possibility that landslides along tracks are more influenced by flood processes than by rainfall alone? This could potentially be attributed to processes such as slope undercutting or toe removal by floodwaters. In times these floods might be caused by a rain event upstream that is spatially not overlapping with the landslide location. Further exploration of this hypothesis could provide valuable insights.

**Figure 2** --> It would be beneficial if the authors could provide a more detailed explanation of how the intersection depicted in this figure was calculated. This aspect seems crucial and might be more appropriately discussed in the results section to enhance the reader's understanding of the methodology. If it is discussed later than the figure's location, it might be an option to add a sentence summary in the caption.

**Figure 3** --> Currently, I find this figure somewhat challenging to interpret. Might the authors consider revising it to enhance its clarity and accessibility? Providing additional context or simplifying the presentation could improve its comprehensibility and overall impact on the reader's understanding.

---

## Author Comment (AC2)

**Reply on RC1**

Introduction:

1. - L52-53 The introduction lacks some literature that links precipitation to tree falls (this linkage might be obvious for gravitational mass movements and floods). Does the year 2021 really proofs this relationship? Isn't it likely that the heavy rain was accompanied by wind gusts? *There are only limited number of studies on precipitation induced tree fall events available. This is mentioned in the discussion section (lines 419-421). If the editor wish, we can add the literature to the introduction section. We agree with the reviewer that heavy precipitation events are often accompaigned by wind gusts so it is difficult to separate clearly what the cause of the tree fall event was. Nevertheless, the influence of soil moisture on the stability and vitality of trees has been proven in various studies (e.g. Usbeck et al. 2010; Hanewinkel et al. 2011; Lucia et al. 2018; WindtrhowHandbook for British Coulmbia Forests 1994).*

Datasets:

1. - L79 spatial intersection has not been explained yet. At least a reference to the section where it will be explained is needed. *We can add a paragraph describing the spatial intersection in detail in the revised manuscript: "The spatial and temporal intersection of the heavy rainfall events and damage events along the railway was carried out by intersecting the CatRaRE polygon data provided by the DWD and the compiled railway damage databases. First, the spatial intersection was carried out using the GIS software ArcMap, version 10.8.1. In ArcMap, the respective damage events tree fall, gravitational mass movement and flooding, which are available as point information, were intersected with the CatRare heavy rainfall events (W3-catalogue) between 2011 and 2021, which are available as area polygons, using the tool "Spatial Join". In the processing options of the Spatial Join tool, the connection process "Join one to many" must be selected so that multiple join features (heavy precipitation events) can be assigned to each target feature (damage event). This creates a database in which all spatially overlapping heavy precipitation events are assigned to the damage events."*

2. - L93-100 spatial intersection should be explained first. This paragraph should be moved into the results section. It is no longer a description of CatRaRE. *We can move this section into the results section.*

3. - L136 Why 5km resolution and not 1km (HYRAS-DE-PRE)? Are you aware that daily precipitation in HYRAS is aggregated from 06 UTC to 06 UTC and that a clear assignment to a single date is not possible? Is daily precipitation the explanatory control variable or 30-day antecedent precipitation (L193), or both? A list of all explanatory control variables is needed in this section (not only a description of the raw data that was used to derive them), otherwise it becomes confusing. *We will add a table with descriptions of all explanatory control variables (see reply to comment 5.). The resolution of the HYRAS data used was 1 km x 1 km, unfortunately the stated resolution of 5km x 5km came from a mixup in the documentation of datasets tested and used. With regard to the time period aggregated we assumed that even despite the time discrepancy the precipitation is still representative for 75% of the day in question. To confront this problem, a variable that represent longer time periods before the event (30-day antecedent precipitation) where also included in the analysis as explanatory control variable.*

4. - L139 DWD soil moisture is based on observations and a soil moisture model. *We will add the information that DWD soil moisture is based on observations and a soil moisture model.*
5. - In general, I find it difficult to keep track/distinguish between the different terms (e.g. rain and rail events). A complete list or table in this section on how the terms are used and what they encompass would be helpful (event, natural hazard, observation, explanatory control variables). *In the revised version we will add a table explaining all the important terms. The table will look like this:*

| Heavy rainfall event | Warning Level W3 (events with 25-40 l/m² in 1 hour or 35-60 l/m² in 6 hours) |
| --- | --- |
| Rail / damage events (tree falls, gravitational mass movements, flood events) | Damages recordes by DB Netz AG; resulting from a damage database of DB Netz AG |
| Damage database | Damage database of DB Netz AG in which the damage events along the railway lines are listed |
| Natural hazard | In this context: tree falls, gravitational mass movements, flood events |
| Natural hazard event datasets | Individual datasets of each natural hazard resulting from the damage database of DB Netz AG |
| Track section (railway) | Defined by the GIS-layer "geo-strecke" provided by DB Netz AG. According to this layer, the German rail netork is divided into 15939 track sections. |
| Route segment (railway) | A section of the German rail network between two adjacent operating points. The total length of the German rail network owned by DB is 56939 of tracks km, which is divided into 9679 route segments. The segments differ in length between 140 m and 12.7 km with an average length of 3.4 km. |
| Explanatory control variables | Climatological and hydrometeorological variables related to the investigated natural hazards to check for other relationships in the statistical regression analysis. Variables used in this study are: daily precipitation, daily soil moisture and hazard indication map for slope and embankment landslides. |
| Observation | Description within the statistical analysis methods for the size of the data set as a combination of days and route segments. The complete dataset is available for 3987 days (= time-series units) and 9679 route segments, resulting in a total of 38590173 route segment - day combinations. The number of observations used in the succeeding models vary depending on the available time period of the natural hazard event datasets. |

Methods:

1. - L156 I am not convinced. A flood can be the result of long-lasting precipitation that is not categorized as heavy (as can be seen this year in Germany and Britain). *We agree that long-lasting precipitation not categorized as heavy can trigger a flood. But it is not the scope of the sentence to describe triggers for flood events. The scope of the sentence is to define which heavy precipitation events will be regarded as possible trigger for secondary processes in this study.*

2. - L166 Please explain the analysis of natural breaks (in which data set?) and its results and how this supports the choice of the time period. *The paper already mentions that this methodology is used because there is no other scienitfically valid source on threshold values when precipitation can have an impact on the natural hazeads analysed. In the revised version, we can add an explanation and a figure on the natural breaks, either in the text or in the appendix, something like this: „This can be observed in Figure XY. In this graph, the days between the occurrence of the damage event and the heavy rainfall event are plotted on the Y-axis and the number of overlapping events for all three types of damage are plotted on the X-axis. The black line after 3 days difference marks the limit used in our analyses and clearly visible in the data, up to which a connection between heavy rain and damage event can be established. It is well visible that for all event types up to day 3, a large number of damage events can be linked to a heavy rain event that occurred immediately before. From day 4 onwards, this link decreases considerably, so that one can consider a natural break in the data, so that the heavy rain events that occurred more than 3 days before a damage event can no longer be clearly identified as the cause for the occurrence of the event. Due to this natural break in the data, which can be observed almost identically for all event types, this limit value of the difference of 3 days between the onset of a heavy rain event and the damage event is used in the further analyses."*

[Figure]

3. - L171 Corresponding to what? *„Corresponding" refers to the values from the explanatory control variables.*

4. - L174 Please explain what panel data analysis, cross sectional analysis and random-effects logistic regression are, for what kind of investigations they should be used and why you chose to apply them here. As these methods are mostly used in social and economic science. It shouldn't be assumed that the methods are known by the audience this article addresses (geological/physical/climatological scientists). I suggest to also write down the equations, explain all variables and terms in the equations using an example from this study. *We would suggest adding the following explanation at the beginning of the sections 2.2.3.1 and 2.2.3.2,*

*respectively:*

*„For the evaluation of the datasets, panel data analysis was chosen as statistical method as it allows the analysis of two- or multidimensional panel data by running regression models over chosen dimensions. The method, originating in the econometrics, is used to observe and explore the relationships between observations of heavy rainfall and natural hazard damage events as the relationships may be very complex and the aim is to explore them further. The dimensions of the data collected for panel data analysis are typically covering the temporal and spatial dimension, here they are time and route segment with or without an event/observation as well as additional explanatory variables that take into account the heterogeneity of the studied individuals. The panel dataset has a matrix structure and includes observations and explanatory variables for each individual route segment for each day of the studied time period (Biørn 2017). The individual or in this case the route segment can be observed over a long time period and opposed to time-series and cross-section data, the effects of individual-specific variables as well as time-specific variables can be explored in a panel analysis. A typical panel data regression model is represented by the equation $y = \beta_0 + \beta_1 x_{ij} + \beta_2 \varepsilon$, where y ist the dependent variable, x is the independent variable, $\beta_0$, $\beta_1$ and $\beta_2$ are coefficients and $\varepsilon$ is an explanatory control variable, i and j are indices for the dimensions chosen for the analysis (e.g. time and space/individual). The limits of panel data analysis are determined by the data quality and consistency, distortions of measurement errors, short time-series dimensions as well as the relationship between the variables due to potential variable bias and unobserved nuisance variables which are correlated to the observable explanatory variables in the equation (Baltagi 2005; Biørn 2017).*

*In this analysis, the authors were interested especially in the probability of which the natural hazard damage events occur in relation to the heavy rainfall events. This was modeled by employing a logit link function, the natural log of the probability that a natural hazard event occurs divided by the probability that it does not occur. Non-linear models are in this case more suitable for modeling binary responses (Wooldridge 2010). To account for the unobserved individual heterogeneity and characteristics (e.g. small scale topography and vegetation) of each route segment, where the damage and heavy rainfall events can occur, a random variable was introduced and a mixed random-effects model used for the estimation of the parameters.“*

*„A cross-sectional analysis is employing a similar equation and logistic model as the panel analysis, but it aims at exploring the effects of one independent variable upon a dependent variable of interest at a certain point in time. This is done by using econometric methods to effectively hold other factors fixed. This approach is limited when the control variables are not completely considered and not measured with the same quality (Wooldridge 2010).“*

5. Please homogenize the usage of indices. i is used for route segments, time lags and combinations of route segments with heavy rain events. This makes it difficult to understand the equations. *We understand that the usage of i as the index of choice in all equations may be confusing. We therefore suggest to replace the index for the number of days after heavy rainfall with „d“ in equation 2, the route segment index with „r“ in equation 3a and 3b, the index for observations of heavy rainfall events in equation 4 with „e“.*

6. - L182 Single point in time (which point in time?) or rather all event time steps together? Please explain in more detail. *This sentence refers to the cross sectional analysis which indeed is only calculated considering one point in time. We explain this further in the answer to question 4.*

7. - L185 The route segment description would better fit in the datasets section. *We can remove the route segment description in the dataset section if the editor wishs. However, we find it most suitable in this chapter because it is the first time where the term route segment is used and it is needed to understand what is done in the panel data analysis.*

8. - L200 Why location of beginning of segment and not the middle? *The segment id and closest station and stop („Betriebsstelle") at the beginning of the segments was the most accurate and complete information about the localisation of the damage events in the damage database. Due to the heterogeneic shapes and kilometrage of segments, more detailed localisation might have introduced more sources of error and thus abandoned.*

9. - L212 This is a normal logistic regression approach (not random-effects). There seems to be a lot of correlation between the independent variables (rain amount and heavy precipitation event, antecedent precipitation index, 30-day precipitation and soil moisture, topographic index and hazard zone ....). Is that a problem for your analysis? What are the consequences for the interpretation of the results? I assume that the OR for these variables is underestimated because parts of the effect are captured by the correlated variables. *The random variable and random effects model was indeed introduced later in equation 3 a and 3b, the results are shown in table 2 and 3. The logistic regression approach and cross-sectional logit model approach is shown in equation 4, with the results in table 4. The various parameters were all investigated despite carrying information about very similar triggers for damage events to give more insight into the magnitude of the relationship between measurements or indicators and different damage events. A larger problem may be the different scales of the variable units when compared against each other.*

10. - L231 I do not understand the reasoning behind the approach to include annual and seasonal dummies. Please explain it in terms of physical mechanisms. The annual and seasonal variability is already captured by the inclusion of the precipitation events and the precipitation amounts that also have a seasonal cycle and annual variability. What additional processes do the annual and seasonal dummies represent? Please also explain that dummy means binary. Suggestion: If you need to capture an annual cycle a good approach are harmonic functions (e.g. https://doi.org/10.1016/j.spasta.2017.11.007) *The seasonal and annual effects may also lead to characteristics such as varying temperature uncoupled from precipitation, resulting soil infiltration capacity, vitality of the vegetation, foilage coverage, onset of the vegetation period, distribution of storms without especially heavy precipitation etc. that were not captured by the input data. We will also add that the dummy variables are binary.*

11. - L237-238 (Eq 3a and 3b) To me it seems that the method you apply is actually a mixed-effects logistic regression model with random effects (mu_i) and fixed effects (all other effects). I would interpret mu_i as a constant offset that affects the mean probability and depends on the rail segment. Without it the equation has the form of an ordinary logistic regression model. I don't understand the idea behind this approach. What is/are the physical characteristics that differ between segments but are constant within a segment. It seems you already included the relevant geological information as geological control variables (e.g. hazard class, topographic information ...). *After looking further into the definition of models, we agree that due to the fixed-term effects, the model should be considered a mixed model with fixed and random effects. As correctly stated in the comment and described in the article, the component $\mu\_i$ is individual-specific to the route segment. It was necessary to introduce this variable as during the development and testing of the panel analysis not all*

*route segment characteristics could be presented by the hazard class and topographic information. We suspect that the input data resolution is in some cases not sufficient to characterize the route segments in detail, may it be due to the smallscale location of the tracks on a slightly elevated dam, the actual exposition, other mitigating factors such as drainage ditches etc. We are welcome to expand this discussion in the revised version.*

12. - L245: Please explain from a physical point of view why you investigate these interactions (and why others are not studied). If you include interaction terms more than one regression coefficient is relevant for the variable. I have doubts that you can use the OR (L219) calculated from just one coefficient to compare the importance of the independent variables if you include interaction terms. *From other studies that have been mentioned in section 2.2.1, rainfall events could be a major trigger for different natural hazards. In the initial exploratory analysis of the relationship between heavy rainfall event represented simply by precipitation the day of and damage events, a limited correlation and a certain time lag could be observed. The additional interaction terms were introduced to account for possible preexisting soil water calculated solely based on the precipitation or also considering evapotransporation.*

13. - Table 1 Why don't you create one table including all variables included in the model for a better overview (soil moisture, seasonal dummy, rail segment, 30-day precipitation, .... ). Please also extend the description. What is meant by specified duration? What is the topographic position index? *We are happy to expand the explanations of the variables used in the mixed-effect regression model with random effects:*

| | Variable | Description |
|---|---|---|
| **Dummy variables** | 0 to 2 days from heavy rainfall event | Binary dummy variable that describes whether the damage event took place between the day of a heavy rainfall event or up to two days after |
| | - day of heavy rainfall | Binary dummy variable that describes whether the damage event took place on the day of a heavy rainfall event |
| | - 1 day after heavy rainfall | Binary dummy variable that describes whether the damage event took place one day after a heavy rainfall event |
| | - 2 days after heavy rainfall | Binary dummy variable that describes whether the damage event took place two days after a heavy rainfall event |
| **Control variables** | Precipitation at route segment [mm] | Daily precipitation on the day of the damage event from the 1 km x 1 km HYRAS dataset |
| | Accumulated precipitation at route segment, 30 days [mm] | 30-days antecedent precipitation calculated based on daily precipitation from the 1 km x 1 km HYRAS dataset |
| | Daily soil moisture at route segment [% nFK] | Daily soil moisture on the day of the damage event from DWD soil moisture 1 km x 1 km grid for agrometeoroligcal applications |

*.... and in the cross-sectional logit model:*

| Abbreviation | Description |
|---|---|
| H | Duration [h] of the heavy rainfall event |
| RRmean | Mean precipitation [mm] of all RADKLIM pixels within the event zone |
| SRImean | Mean of the heavy rainfall index (in German "Starkregenindex") within the event zone: An index describing the speed at which rainfall accumulates within a specified duration of time. Mean of all RADKLIM-pixels within the event zone (Range [0,12]) |
| V3_AVG | Mean of the 21-days antecedent precipitation index within the event zone |
| ETA | A measure of the extremity of the heavy rain event as a function of the return period as well as |

| | affected area of an event |
|---|---|
| VSGL_GRAD | Mean degree of sealing [%]: Percentage of sealed area including road infrastructure within the event zone |
| STRM_AVG | Mean elevation [m] above sea level within the event zone |
| TPI_AVG | Mean of the Topographic Position Index that classifies the landscape into upper, middle and lower parts, 2 km circular neighborhood, in the event zone within Germany |

Results:

1. - L284 Does this also hold for the individual processes. Fig. 3 only shows the combined result. *These results also hold for the individual processes as mentioned in the Figure caption: „All three natural hazard processes are shown together in the figure as the distribution looks similar for each process when viewed individually."*

2. - L296 ...the higher the value... (of log likelihood or sample size?) *„The higher the value" refers to the log likelihood.*

3. - L297 The AIC is useful for comparing models of different complexity. However, you can only compare the AIC if the models are fitted using the same number of observations. This is not the case here. *The AIC shows that the models based on the two different equations (1) and (2) offer a similar quality. Overall the quality of the results is rather low as the number/proportion of events in the panel dataset is small compared to the whole dataset.*

4. - Table 2 and Table 4 Please give the full model equations for the 3 hazards. Do you use the same equation for all 3 hazards (e.g. is the hazard indication map for slope and embankment landslides used for all 3 hazards)? How many parameters did you have to fit for each hazard model? As each segment needs one parameter it must be more than 9679. This is a lot compared to the number of natural hazard events that occurred during the analysis period (14461 trees, 1269 floods, 418 gravitational mass movements). Can you rule out overfitting? *For the panel analysis, the three hazards were fitted/modelled separately to the same equations (1) and (2). For each hazard model, 5 coefficients and one random variable had to be fitted. The small number of observations of damage events is indeed a problem and overfitting cannot be ruled out as the overwhelming majority of observations are without. For the cross-sectional analysis, for each of the three hazards and for each of the investigated variables, a separate model with model coefficients was fitted.,*

1. – L340 indicate … non-linearity … What brings you to this conclusion? *The tables show, at least on the basis of the data and equations applied, that soil moisture has a significant effect on the probability of natural hazards occurring. The analysis shows that a higher soil moisture has a slightly negative effect on the probability in the analysis not broken down by individual days (Table 2). The analysis broken down by days between heavy rainfall and damage events shows a slight positive influence (Table 3). As both results are significant, this could either be a false correlation or an indication of non-linear effects. The latter is also suggested by the graphs in Fig. 4.*

2. - L344 The analysis considers only the interaction terms? Please elaborate what you have done. *The analysis shown in figure 4 uses the equation 3a) with the coefficients calculated for the different natural hazards. For the available range of the control variables, the probabilities predicted/calculated with the three different models is shown.*

3. - Figure 4 What does the figure show? The probability or a prediction of the probability using the statistical model? Panels a,d,e: If precipitation becomes strong enough the day should

automatically become a heavy rainfall event. Why are there two different curves at high precipitation values? Panels c,f,i: Do these curves make any sense in terms of physics? I assume they are a statistical model artifact. Soil moisture was one of the non-significant parameters. *The first part of this comment has been adressed in the answer above. Panels a,d,e: This is correct. With higher precipitation, the curves should be the same, as there would no longer be any days without a heavy rainfall event. The definition of heavy rain events in this analysis is, as described, defined by their Warning Level W3 (events with 25-40 l/m² in 1 hour or 35-60 l/m² in 6 hours). Panels c.f.i: The curves do make sense in terms of physics. In panel c, the probability of a flood is exactly the opposite to the soil infiltration capacity depending on the soil moisture content: water infiltration is low when the soil is especially dry and the pores are closed as well as when they are overfull. In panel f: not significant, but high water infiltration may lead to slope destabilization. In panel i: At very low soil moisture, trees suffer from drought stress and have low vitality. If a heavy rainfall event occurs, the unhealthy trees may be more likely to fall. A similar situation can occur with high soil moisture. The soil is already softened/weak and saturated. In the event of a heavy rainfall event, this can lead to faster falling.*

4. - L384 Could this mean that there are no trees if the soil is sealed and therefore the probability for tree fall is low? *Yes, this can be a possible explanation.*

Technical corrections

1. - L84/85 Is there a difference between events and heavy rainfall events? *There is no difference. For more clarity, we can rewrite the sentence: „In addition, many heavy rainfall events occurred in May and September, while they were rare during winter."*
2. - L177 proximity in space or time? *Proximity in time is meant here.*
3. - L212 is the prime symbol missing from beta2? *Yes, we will add it in the revised version.*
4. - L264 is the prime symbol missing for beta? *Yes, we will add it in the revised version.*

---

## Author Response (AR1)

**Responses to the editor decision**
* * *
Response to RC1:

Introduction, Lines 52-53: It should become clear that treefall events can be caused without any precipitation event, and that there is literature pointing at an enhanced treefall risk with storms and (previous) precipitation, i.e. enhanced soil moisture. I agree with the reviewer that the "proof" of the relationship by pointing at the event of summer 2021 is not convincing.

*In the revised version, we added more literature that show the influence of soil moisture on the stability and vitality (Hanewinkel et al., 2011; Lucia et al., 2018; Usbeck et al., 2010) and rewrote the sentence about the event of summer 2021, so that it becomes clear, that this event is not a „proof" of the relationship. For the complete response see document „Reply on RC1".*

Methods:

1. Please clarify in the manuscript that you try to assign flood events to local rainfall extremes occurring in a short time frame before the effect of flooding on the railroad.

*We added some explanatory sentences at the beginning of the paragraph: „Although long-lasting precipitation not categorized as heavy can also trigger floods and can have several impacts on railway infrastructure, the scope of this study is to try to assign flood events to local rainfall extremes occurring in a short time frame before they effect the railway system. So in the context of this study, a heavy rainfall event was only be considered as a trigger for a secondary process if the heavy rainfall event occurs directly or shortly before the damage event."*

7. Please move the explanation as suggested. The reader should have a concise and compact description of what you did.

*The route segment description is now given in the new Table 1 at the beginning of chapter 2.1.*
* * *
Reviewer 3

I agree with Reviewer 3 that it would be good for readers to have a better understanding of the dataset limitations. Thus, I suggest to add short statements:

1.1 provide the basic information (not all details) of the general keywords used to distinguish between the hazards, and give some information on what percentage of non-matches you have. If a permission is needed, you can ask benjamin.schmitz@deutschebahn.com. He told me he would ask the responsible people about your suggested additions to the revised version of your paper.

*The responses given in „Reply on RC3" and the new Figure 3 were coordinated with Benjamin Schmitz: „The authors are not part of the DB Group, and the DB Netz AG database is an internal documentation that is not publicly available. Unfortunately, we have no rights to publish raw data from the database (it includes any disruption within the railway system) and only receive extracts from the database (as .csv) and have no access to the database itself. We are therefore unfortunately unable to show raw data for data protection reasons. In the revised version, we added some more descriptions on the dataset (chapters 2.1.2 and 4.2) and included a new figure showing one example for each type of natural hazard with a picture and the information given on the example event in the database. As the information on the processes itself are rather limited, the event information does not differ between events that match with rainfall events and those which not match. Therefore we show only one example per natural hazard process. We decided not to explicitly mention again the percentages of matches vs. non matches as these numbers are already mentioned as major results in the manuscript (e.g. lines 10-11)."*

1.2 I guess this event is not an example for a non-match, so I think there is no need to react.
1.3 And 2.: Apparently, it should be made clearer that a flood can be produced by upstream rainfall rather than by local rainfall, and this rainfall may not reach local extremes thresholds but could be wide-spread in order to produce flooding. This clarification is demanded by all reviewers!
*We gave detailed responses to the comments 1.3, 2.1-2.4 of RC3 and addresed this point in the discussion (chapter 4.2) in more detail and also in the description of the dataset (chapter 2.1.2) in the revised version.*

Minor comments:
I would like to allow explicitly to have a few words more than the 199 letter limit, so that you can include the hazard types in the abstract.
*We included the hazard types in the abstract as suggested.*
* * *
Reviewer 2:

Please mention the source of the 21-day choice for antecedent rainfall.
*We added an explanation why we chose the 21-day antecent rainfall in chapter 4.1.*
Line 100 comment: Indeed, readers outside Germany may not be able to identify states. You might consider just to put a light background color on the left map to the state mentioned in the text.
*We reworked Figure 2 in the way that the location of the federal states mentioned in the manuscript is marked in map (a).*
Line 153: Please mention that there are very few cases with multiple rainfall events associated to one natural hazard event.
*There are only very few cases with multiple rainfall events associated to one natural hazard event in or dataset. We added some explanatory sentences on this issue at the end of the following paragraph in the revised version.*
Line 325: Please clarify the rarity.
*We added some explanatory sentences to clarify the rarity in the revised version.*
* * *
Response to CC2:
Obviously, the relation of absolute changes to change rates (in %) of the different quantities are not clear. Please add.
*Please can you specify to which of the comments on CC2 your comment refers, so that we can answer the question correctly?*

**Reply on RC1**

Introduction:

1. - L52-53 The introduction lacks some literature that links precipitation to tree falls (this linkage might be obvious for gravitational mass movements and floods). Does the year 2021 really proofs this relationship? Isn't it likely that the heavy rain was accompanied by wind gusts? *There are only limited number of studies on precipitation induced tree fall events available. This is mentioned in the discussion section (lines 419-421). We agree with the reviewer that heavy precipitation events are often accompaigned by wind gusts so it is difficult to separate clearly what the cause of the tree fall event was. Nevertheless, the*

*influence of soil moisture on the stability and vitality of trees has been proven in various studies (e.g. Usbeck et al. 2010; Hanewinkel et al. 2011; Lucia et al. 2018). We have rewritten the section and it now reads as follows: „These events can be triggered by a variety or a combination of different factors, but heavy rainfall events are possible triggers for all of these processes. This relationship is much clearer for floods and gravitational mass movement processes, while it is not so straightforward for tree falls. There are only a limited number of studies on precipitation induced tree fall events available (e.g. Morimoto et al. 2021), and heavy precipitation events such as the event in July 2021 are often accompanied by wind gusts, so it is difficult to separate clearly what the exact cause of the tree fall event was. However, the influence of soil moisture on the stability and vitality of trees has been proven in various studies (e.g. Hanewinkel et al., 2011; Lucia et al., 2018; Usbeck et al., 2010)."*

Datasets:

1. - L79 spatial intersection has not been explained yet. At least a reference to the section where it will be explained is needed.
*We have added a short description to the manuscript on the question of the spatial intersection of the CatRaRE data with the rail network.*
*In fact, this is a simple spatial intersection, i.e. a comparison of which heavy rain events (polygons from CatRaRE) affect the rail network. No buffers, possible drainage areas or other factors were included.*
2. - L93-100 spatial intersection should be explained first. This paragraph should be moved into the results section. It is no longer a description of CatRaRE. *We would like to leave this paragraph here and not move it to the results, as we have separated it in the way that the description of the datasets and the first analyses are given in the „Methods" and the intersection of the two datasets and the modelling results in the „Results". However, it is clear to us that methods and results are not completely separated but we found it more straightforward this way, as the first analyses gives a good impression of the datasets and contribute to a better understanding of the following intersection and modelling, but they are not the core topic of the manuscript.*
3. - L136 Why 5km resolution and not 1km (HYRAS-DE-PRE)? Are you aware that daily precipitation in HYRAS is aggregated from 06 UTC to 06 UTC and that a clear assignment to a single date is not possible? Is daily precipitation the explanatory control variable or 30-day antecedent precipitation (L193), or both? A list of all explanatory control variables is needed in this section (not only a description of the raw data that was used to derive them), otherwise it becomes confusing. *We will add a table with descriptions of all explantory control variables (see reply to comment 5.). The resolution of the HYRAS data used was 1 km x 1 km, unfortunately the stated resolution of 5km x 5km came from a mixup in the documentation of datasets tested and used. With regard to the time period aggregated we assumed that even despite the time discrepancy the precipitation is still representative for 75% of the day in question. To confront this problem, a variable that represent longer time periods before the event (30-day antecedent precipitation) where also included in the analysis as explanatory control variable.*
4. - L139 DWD soil moisture is based on observations and a soil moisture model. *We will add the information that DWD soil moisture is based on observations and a soil moisture model.*
5. - In general, I find it difficult to keep track/distinguish between the different terms (e.g. rain and rail events). A complete list or table in this section on how the terms are used and what

they encompass would be helpful (event, natural hazard, observation, explanatory control variables). *In the revised version we added a table explaining all the important terms at the beginning of chapter 2.*

| Term | Description |
|---|---|
| Heavy rainfall event | Warning level W3 (events with 25-40 l/m² in 1 hour or 35-60 l/m² in 6 hours) |
| Rail / damage events (tree falls, gravitational mass movements, flood events) | Damages recorded by DB Netz AG; resulting from a damage database of DB Netz AG |
| Damage database | Damage database of DB Netz AG in which the damage events along the railway lines are listed |
| Natural hazard | In this context: tree falls, gravitational mass movements, flood events |
| Natural hazard event datasets | Individual datasets of each natural hazard resulting from the damage database of DB Netz AG |
| Track section (railway) | Defined by the GIS-layer "geo-strecke" provided by DB Netz AG. According to this layer, the German rail netork is divided into 15939 track sections. |
| Route segment (railway) | A section of the German rail network between two adjacent operating points. The total length of the German rail network owned by DB is 56939 of tracks km, which is divided into 9679 route segments. The segments differ in length between 140 m and 12.7 km with an average length of 3.4 km. |
| Explanatory control variables | Climatological and hydrometeorological variables related to the investigated natural hazards to check for other relationships in the statistical regression analysis. Variables used in this study are: **daily precipitation**, **daily soil moisture** and **hazard indication map for slope and embankment landslides**. |
| Observation | Description within the statistical analysis methods for the size of the dataset as a combination of days and route segments.
The complete dataset is available for 3987 days (= time-series units) and 9679 route segments, resulting in a total of 38590173 route segment - day combinations.
The number of observations used in the succeeding models vary depending on the available time period of the natural hazard event datasets. |

Methods:

1. - L156 I am not convinced. A flood can be the result of long-lasting precipitation that is not categorized as heavy (as can be seen this year in Germany and Britain). *We agree that long-lasting precipitation not categorized as heavy can trigger a flood. But it is not the scope of the sentence to describe triggers for flood events. The scope of the sentence is to define which heavy precipitation events will be regarded as possible trigger for secondary processes in this study. To clarify this context, we rewrote the beginning of the paragraph as follows: „Although long-lasting precipitation not categorized as heavy can also trigger floods and can have several impacts on railway infrastructure, the scope of this study is to try to assign flood events to local rainfall extremes occurring in a short time frame before they effect the railway system. So in the context of this study, a heavy rainfall event was only be considered as a trigger for a secondary process if the heavy rainfall event occurs directly or shortly before the damage event."*
2. - L166 Please explain the analysis of natural breaks (in which data set?) and its results and how this supports the choice of the time period. *The paper already mentions that this methodology is used because there is no other scienitfically valid source on threshold values when precipitation can have an impact on the natural hazards analysed. In the revised version, we added a explanatory text: "By plotting the days between the occurrence of the damage event and the heavy rainfall event against the number of overlapping events, it is well visible that for all three types of natural hazards up to day 3 a large number of damage events can be linked to a heavy rainfall event that occurred immediately before. From day 4*

*onwards, this link decreases considerably, so that the heavy rainfall events occurring more than three days before a damage event can no longer be clearly identified as cause for the occurrence of the event. Due to this natural break in the data, the limit value of the difference of three days between the onset of a heavy rainfall event and the damage event was used in the further analyses."*

3. - L171 Corresponding to what? *„Corresponding" refers to the values from the explanatory control variables.*
4. - L174 Please explain what panel data analysis, cross sectional analysis and random-effects logistic regression are, for what kind of investigations they should be used and why you chose to apply them here. As these methods are mostly used in social and economic science. It shouldn't be assumed that the methods are known by the audience this article addresses (geological/physical/climatological scientists). I suggest to also write down the equations, explain all variables and terms in the equations using an example from this study. *We added the following more detailed explanation at the beginning of chapter 2.2.3:*
   *Panel analysis: „For the evaluation of the datasets, panel data analysis was chosen as statistical method as it allows the analysis of two- or multidimensional panel data by running regression models over chosen dimensions. The method, originating in the econometrics, is used to observe and explore the relationships between observations of heavy rainfall and natural hazard damage events as the relationships may be very complex and the aim is to explore them further. The dimensions of the data collected for panel data analysis are typically covering the temporal and spatial dimension, here they are time and route segment with or without an event/observation as well as additional explanatory variables that take into account the heterogeneity of the studied individuals. The panel dataset has a matrix structure and includes observations and explanatory variables for each individual route segment for each day of the studied time period (Biørn 2017). The individual or in this case the route segment can be observed over a long time period and opposed to time-series and cross-section data, the effects of individual-specific variables as well as time-specific variables can be explored in a panel analysis. A typical panel data regression model is represented by the equation $y = \beta_0 + \beta_1 x_{ij} + \beta_2 \varepsilon$, where y ist the dependent variable, x is the independent variable, $\beta_0$, $\beta_1$ and $\beta_2$ are coefficients and ε is an explanatory control variable, i and j are indices for the dimensions chosen for the analysis (e.g. time and space/individual). The limits of panel data analysis are determined by the data quality and consistency, distortions of measurement errors, short time-series dimensions as well as the relationship between the variables due to potential variable bias and unobserved nuisance variables which are correlated to the observable explanatory variables in the equation (Baltagi 2005; Biørn 2017). In this analysis, the authors were interested especially in the probability of which the natural hazard damage events occur in relation to the heavy rainfall events. This was modeled by employing a logit link function, the natural log of the probability that a natural hazard event occurs divided by the probability that it does not occur. Non-linear models are in this case more suitable for modeling binary responses (Wooldridge 2010). To account for the unobserved individual heterogeneity and characteristics (e.g. small scale topography and vegetation) of each route segment, where the damage and heavy rainfall events can occur, a random variable was introduced and a mixed random-effects model used for the estimation of the parameters."*
   *Cross-sectional analysis: „A cross-sectional analysis is employing a similar equation and logistic model as the panel analysis, but it aims at exploring the effects of one independent*

*variable upon a dependent variable of interest at a certain point in time. This is done by using econometric methods to effectively hold other factors fixed. This approach is limited when the control variables are not completely considered and not measured with the same quality (Wooldridge 2010).“*

5. Please homogenize the usage of indices. i is used for route segments, time lags and combinations of route segments with heavy rain events. This makes it difficult to understand the equations. *We understand that the usage of i as the index of choice in all equations may be confusing. We therefore suggest to replace the index for the number of days after heavy rainfall with „d“ in equation 2, the route segment index with „r“ in equation 3a and 3b, the index for observations of heavy rainfall events in equation 4 with „e“.*
6. - L182 Single point in time (which point in time?) or rather all event time steps together? Please explain in more detail. *This sentence refers to the cross sectional analysis which indeed is only calculated considering one point in time. We explain this further in the answer to question 4, and the explanation is integrated in the revised version in section 2.2.3.*
7. - L185 The route segment description would better fit in the datasets section. *The route segment descripition is now given in the new Table 1 at the beginning of chapter 2.1. However, we find it necessary to give the explanation again in this line (with reference to Table 1) for a better understanding of what is done in the panel data analysis.*
8. - L200 Why location of beginning of segment and not the middle? *The segment id and closest station and stop („Betriebsstelle“) at the beginning of the segments was the most accurate and complete information about the localisation of the damage events in the damage database. Due to the heterogeneic shapes and kilometrage of segments, more detailed localisation might have introduced more sources of error and thus abandoned. We added an explanatory sentence in the revised manuscript.*
9. - L212 This is a normal logistic regression approach (not random-effects). There seems to be a lot of correlation between the independent variables (rain amount and heavy precipitation event, antecedent precipitation index, 30-day precipitation and soil moisture, topographic index and hazard zone ....). Is that a problem for your analysis? What are the consequences for the interpretation of the results? I assume that the OR for these variables is underestimated because parts of the effect are captured by the correlated variables. *The random variable and random effects model was indeed introduced later in equation 3 a and 3b, the results are shown in table 2 and 3. The logistic regression approach and cross-sectional logit model approach is shown in equation 4, with the results in table 4. The various parameters were all investigated despite carrying information about very similar triggers for damage events to give more insight into the magnitude of the relationship between measurements or indicators and different damage events. A larger problem may be the different scales of the variable units when compared against each other.*
10. - L231 I do not understand the reasoning behind the approach to include annual and seasonal dummies. Please explain it in terms of physical mechanisms. The annual and seasonal variability is already captured by the inclusion of the precipitation events and the precipitation amounts that also have a seasonal cycle and annual variability. What additional processes do the annual and seasonal dummies represent? Please also explain that dummy means binary. Suggestion: If you need to capture an annual cycle a good approach are harmonic functions (e.g. https://doi.org/10.1016/j.spasta.2017.11.007) *The seasonal and annual effects may also lead to characteristics such as varying temperature uncoupled from precipitation, resulting soil infiltration capacity, vitality of the vegetation, foilage coverage,*

*onset of the vegetation period, distribution of storms without especially heavy precipitation etc. that were not captured by the input data. We added this sentence and also mentioned that the dummy variables are binary.*

11. - L237-238 (Eq 3a and 3b) To me it seems that the method you apply is actually a mixed-effects logistic regression model with random effects (mu_i) and fixed effects (all other effects). I would interpret mu_i as a constant offset that affects the mean probability and depends on the rail segment. Without it the equation has the form of an ordinary logistic regression model. I don't understand the idea behind this approach. What is/are the physical characteristics that differ between segments but are constant within a segment. It seems you already included the relevant geological information as geological control variables (e.g. hazard class, topographic information ...). *After looking further into the definition of models, we agree that due to the fixed-term effects, the model should be considered a mixed model with fixed and random effects. As correctly stated in the comment and described in the article, the component µ_i is individual-specific to the route segment. It was necessary to introduce this variable as during the development and testing of the panel analysis not all route segment characteristics could be presented by the hazard class and topographic information. We suspect that the input data resolution is in some cases not sufficient to characterize the route segments in detail, may it be due to the smallscale location of the tracks on a slightly elevated dam, the actual exposition, other mitigating factors such as drainage ditches etc. We expanded this discussion in the revised version.*
12. - L245: Please explain from a physical point of view why you investigate these interactions (and why others are not studied). If you include interaction terms more than one regression coefficient is relevant for the variable. I have doubts that you can use the OR (L219) calculated from just one coefficient to compare the importance of the independent variables if you include interaction terms. *From other studies that have been mentioned in section 2.2.1, rainfall events could be a major trigger for different natural hazards. In the initial exploratory analysis of the relationship between heavy rainfall event represented simply by precipitation the day of and damage events, a limited correlation and a certain time lag could be observed. The additional interaction terms were introduced to account for possible preexisting soil water calculated solely based on the precipitation or also considering evapotransporation. We added an explanatory sentence in the revised manuscript.*
13. - Table 1 Why don't you create one table including all variables included in the model for a better overview (soil moisture, seasonal dummy, rail segment, 30-day precipitation, .... ). Please also extend the description. What is meant by specified duration? What is the topographic position index? *We are happy to expand the explanations of the variables used in the mixed-effect regression model with random effects:*

| | Abbr. | | Variable | Description |
|---|---|---|---|---|
| Dummy variables | HR | | 0 to 2 days from heavy rainfall event | Binary dummy variable that describes whether the damage event took place between the day of a heavy rainfall event or up to two days after |
| | DHR | $d_0$ | - day of heavy rainfall | Binary dummy variable that describes whether the damage event took place on the day of a heavy rainfall event |
| | | $d_1$ | - 1 day after heavy rainfall | Binary dummy variable that describes whether the damage event took place one day after a heavy rainfall event |
| | | $d_2$ | - 2 days after heavy rainfall | Binary dummy variable that describes whether the damage event took |

| | Abbr. | Variable | Description |
|---|---|---|---|
| | | | place two days after a heavy rainfall event |
| Control variables | dP | Precipitation at route segment [mm] | Daily precipitation on the day of the damage event from the 1 km x 1 km HYRAS dataset |
| | acP | Accumulated precipitation at route segment, 30 days [mm] | 30-days antecedent precipitation calculated based on daily precipitation from the 1 km x 1 km HYRAS dataset |
| | dSM | Daily soil moisture at route segment [% nFK] | Daily soil moisture on the day of the damage event from DWD soil moisture 1 km x 1 km grid for agrometeoroligcal applications |

*…. and in the cross-sectional logit model (selected from the CatRaRe dataset):*

| Abbreviation | Description |
|---|---|
| H | Duration [h] of the heavy rainfall event |
| RRmean | Mean precipitation [mm] of all RADKLIM pixels within the event zone |
| SRImean | Mean of the heavy rainfall index (in German "Starkregenindex") within the event zone: An index describing the speed at which rainfall accumulates within a specified duration of time. Mean of all RADKLIM-pixels within the event zone (Range [0,12]) |
| V3_AVG | Mean of the 21-days antecedent precipitation index within the event zone |
| ETA | A measure of the extremity of the heavy rain event as a function of the return period as well as affected area of an event |
| VSGL_GRAD | Mean degree of sealing [%]: Percentage of sealed area including road infrastructure within the event zone |
| STRM_AVG | Mean elevation [m] above sea level within the event zone |
| TPI_AVG | Mean of the Topographic Position Index that classifies the landscape into upper, middle and lower parts, 2 km circular neighborhood, in the event zone within Germany. The index is calculated as the difference between the height of a cell in the DTM and the average height of all neighbouring cells in the sliding window around this cell. It can approximate the topographical wind exposure of mountain and valley locations. |

Results:

1. - L284 Does this also hold for the individual processes. Fig. 3 only shows the combined result. *These results also hold for the individual processes as mentioned in the Figure caption: „All three natural hazard processes are shown together in the figure as the distribution looks similar for each process when viewed individually." We added a corrseponding sentence.*
2. - L296 ...the higher the value... (of log likelihood or sample size?) *„The higher the value" refers to the log likelihood.*
3. - L297 The AIC is useful for comparing models of different complexity. However, you can only compare the AIC if the models are fitted using the same number of observations. This is not the case here. *We agree that the AIC and the log likelihood values have to be considered in relation to the dataset used. When normalized with the number of observations, the AIC shows that the models based on the two different equations (1) and (2) offer a similar quality. Overall the quality of the results is rather low as the number/proportion of events in the panel dataset is small compared to the whole dataset.*
4. - Table 2 and Table 4 Please give the full model equations for the 3 hazards. Do you use the same equation for all 3 hazards (e.g. is the hazard indication map for slope and embankment landslides used for all 3 hazards)? How many parameters did you have to fit for each hazard model? As each segment needs one parameter it must be more than 9679. This is a lot compared to the number of natural hazard events that occurred during the analysis period

(14461 trees, 1269 floods, 418 gravitational mass movements). Can you rule out overfitting? *For the panel analysis, the three hazards were fitted/modelled separately to the same equations (1) and (2). For each hazard model, 5 coefficients and one random variable had to be fitted. The small number of observations of damage events is indeed a problem and overfitting cannot be ruled out as the overwhelming majority of observations are without. For the cross-sectional analysis, for each of the three hazards and for each of the investigated variables, a separate model with model coefficients was fitted.*

5. – L340 indicate … non-linearity … What brings you to this conclusion? *The tables show, at least on the basis of the data and equations applied, that soil moisture has a significant effect on the probability of natural hazards occurring. The analysis shows that a higher soil moisture has a slightly negative effect on the probability in the analysis not broken down by individual days (Table 2). The analysis broken down by days between heavy rainfall and damage events shows a slight positive influence (Table 3). As both results are significant, this could either be a false correlation or an indication of non-linear effects. The latter is also suggested by the graphs in Fig. 4. We added explanatory sentences.*
6. - L344 The analysis considers only the interaction terms? Please elaborate what you have done. *We added the following explanation: „The analysis used the equation 3a) with the coefficients calculated for the different natural hazards. For the available range of the control variables, the probability calculated with the three different models is shown."*
7. - Figure 4 What does the figure show? The probability or a prediction of the probability using the statistical model? Panels a,d,e: If precipitation becomes strong enough the day should automatically become a heavy rainfall event. Why are there two different curves at high precipitation values? Panels c,f,i: Do these curves make any sense in terms of physics? I assume they are a statistical model artifact. Soil moisture was one of the non-significant parameters. *The first part of this comment has been adressed in the answer above. Panels a,d,e: This is correct. With higher precipitation, the curves should be the same, as there would no longer be any days without a heavy rainfall event. The definition of heavy rain events in this analysis is, as described, defined by their Warning Level W3 (events with 25-40 l/m² in 1 hour or 35-60 l/m² in 6 hours). Panels c.f.i: The curves do make sense in terms of physics. In panel c, the probability of a flood is exactly the opposite to the soil infiltration capacity depending on the soil moisture content: water infiltration is low when the soil is especially dry and the pores are closed as well as when they are overfull. In panel f: not significant, but high water infiltration may lead to slope destabilization. In panel i: At very low soil moisture, trees suffer from drought stress and have low vitality. If a heavy rainfall event occurs, the unhealthy trees may be more likely to fall. A similar situation can occur with high soil moisture. The soil is already softened/weak and saturated. In the event of a heavy rainfall event, this can lead to faster falling. We added an explanatory paragraph in chapter 4.1 in the revised version.*
8. - L384 Could this mean that there are no trees if the soil is sealed and therefore the probability for tree fall is low? *Yes, this can be a possible explanation. We already mentioned this assumption in the discussion (chapter 4.1, lines 444-447).*

Technical corrections

1. - L84/85 Is there a difference between events and heavy rainfall events? *There is no difference. For more clarity, we rewrote the sentence: „In addition, many heavy rainfall events occurred in May and September, while they were rare during winter.“*
2. - L177 proximity in space or time? *Proximity in time is meant here.*
3. - L212 is the prime symbol missing from beta2? *Yes, we will add it in the revised version.*
4. - L264 is the prime symbol missing for beta? *Yes, we will add it in the revised version.*

**Reply on RC2**

My **major concerns** concentrate around the method choices forming the foundation of the current study. I highlight the line number of a piece from the manuscript in quotation marks which is followed by my comments after the sign of -->.

**Line 79:** "spatially intersected with the German rail network" --> Is this intersection achieved considering purely spatial overlap, or are rainfall runoff conditions taken into account as well? For instance, rainfall upstream could potentially impact tracks downstream, even in the absence of local precipitation.
*This intersection is achieved considering purely spatial overlap and does not take rainfall runoff conditions into account. We are aware that this is strong simplification and does not represent the real world. However, this aspect is not the central part of the study; our intention of this simple intersection is only to show that large parts of the German rail network can potentially be affected by heavy rainfall events and that it is therefore necessary to have a closer look on this natural hazard. In the revised version, we point out the limitations of the spatial intersection and point out that a detailed consideration is necessary for recommendations for action, etc.*

**Line 97:** "one heavy rainfall event" --> Could the authors clarify if they are referring to hydrogeomorphological events, including mass-wasting process? I suspect that the tree falls might relate more to wind than rainfall. If the tree fall process is indeed related to wind, it might be beneficial to consider a term that encompasses all three phenomena. Perhaps including wind events as a factor, or alternatively, reconsidering the inclusion of tree fall cases, might provide a more concise, easy-to-explain analysis.
*This paragraph refers only to the spatial intersection of the CatRaRE precipitation events with the rail network and does not take into account the effects of heavy rainfall events such as mass movements or tree fall events. The topic that tree fall events are often caused by wind and that heavy rainfall events are often accompanied by wind gusts is already discussed at various points in the manuscript, e.g.: lines 416/17 and references therein, line 436. In the revised version, we added more sentences and literature on this issue (see response to comment of RC1, Introduction).*

**Line 158:** "e.g. shown for shallow landslides" --> While the observation may hold true for shallow landslides, it's important to note that gravitational mass movements also encompass deep-seated landslides, where the lag time could extend significantly, potentially reaching years. Even excluding these exceptional cases, a lag time of 10-15 days appears realistic, as evidenced by Dille et al. (2022; https://doi.org/10.1038/s41561-022-01073-3). Should the focus be on shallow landslides, it would be helpful if this distinction is made clear. The choice of a 2-day lag for considering landslides raises

some concerns for me, and I kindly suggest revisiting this aspect for a more nuanced discussion. The term "gravitational mass movements" might cover more processes than the authors intended.
*We are aware of the problem that the term „gravitational mass movements" encompasses a broad range of different processes, all of which have very different triggering factors and recurrence times. However, no clear process assignement can be derived from the event data along the rail network, so that unfortunately no further differentiation or adapted models (e.g. longer lag times for deep seated landslides) are possible here. The process diversity of gravitational mass movements and the restriction of the DB damage data base are already addressed in the discussion in the manuscript (chapter 4.2), but we added two more sentences in the discussion in the revised version.*

**Lines 200–202:** "The segment is considered to have been affected by a heavy rainfall event on a given day if a heavy rainfall event from the CatRaRE database has occurred on that day up to a maximum of two days previously." --> As I mentioned in my previous comment, I'm concerned that the proposed time-lag window might not adequately capture the lag time associated with landslides. Later on, in the results (comment below), the authors highlight that only a small fraction of gravitational mass movements were linked to certain rain events. Hence, as previously mentioned, extending this window could offer a more accurate representation of the impact of heavy rainfall on landslide occurrences.
*See response to comment from reviewer RC1, Methods, comment 2. The lag-time for all three processes (floods, tree falls and gravitationl mass movements) is rather short.*

**Line 275:** "a total of 59 events (14 %)" --> I wonder if the correlation might become more pronounced if the lag time were extended to 15 days or more. This adjustment could potentially offer a more comprehensive analysis of the impact of heavy rainfall on these events.
*See response to previous comment.*

**Lines 278-279:** "Of the 14461 tree fall events, a total of 312 (2 %) events can be spatially and temporally linked to a heavy rainfall event." --> This observation might suggest an indirect connection between rainfall and tree falls, potentially implicating other factors such as wind (as discussed by Gardiner et al.; http://dx.doi.org/10.2139/ssrn.4576016) or flooding (Lucia et al., 2018; https://doi.org/10.1016/j.scitotenv.2018.05.186). Further exploration of these factors could enrich the study.
*Thank you for the literature references. The paper from Gardiner et al. is already mentioned in the discussion (line 416), but we can add more detailed information on the factors triggering tree fall events in the revised version. We included these sentences in the Introduction (see response to comment of RC1, Introduction).*

My **minor concerns** primarily revolve around the use of terminology and the occasional absence of detailed explanatory statements that could further enhance the manuscript's readability and comprehension. Clarifying these aspects could improve the overall understandability for the readers. I list the minor comments in the attached file to keep my online comments concise.

**Line 6:** "associated natural hazards" --> Could you please specify which natural hazards are being referred to here?
*We added the three associated natural hazards.*

**Line 8:** "random-effects logistic models" --> At this juncture, I'm finding it challenging to grasp the specifics of this model. Could you possibly elaborate further for clarity?
*A detailed description on the set up of the random-effects logistic model is given from line 207 onwards. For more clarity, we included a more detailed description of the models in the revised version (chapter 2.2.3).*

**Line 8:** "DB Netz AG" --> I'm concerned that this acronym might not be readily understood by a significant portion of the NHESS readership outside of Germany. Could a brief explanation be provided for broader accessibility?
*We replaced the abbreviation „DB Netz AG" by the more general term „a damage database from a German railroad operator".*

**Line 8:** "CatRaRe" --> Similarly, this acronym might not resonate with the wider NHESS audience. A definition could greatly aid in understanding.
*We replaced the abbreviation „CatRaRE catalogue of the German Weather Serivce" by the more general term „a catalogue on heavy rainfall events from the German Weather Service".*

**Lines 10-11:** "Twenty-three percent of the flood events, 14% of the gravitational mass movements and 2% of the tree fall events" --> I'm having a bit of difficulty following these percentages. To clarify, does this imply that 77% of the flood events are not attributed to heavy rainfall?
*Yes, the percentages implies that 77% of the flood events, 86% of the gravitational mass movements and 98% of the tree fall events recorded in the damage database of DB Netz AG could not be attributed to a heavy rainfall event from the CatRaRE dataset.*

**Lines 12-13:** "a heavy rainfall event significantly increases the probability of occurrence of a flood by a factor of 34.29." --> Am I correct in understanding that, according to the authors, there are floods that occur independently of heavy rainfall events?
*Yes, this is correct.*

**Line 12:** "Tree fall" --> I'm struggling to conceptualize how heavy rainfall directly leads to tree fall. Could you provide further insight into this connection?
*To provide further insights into this connection is out of the scope of the abstracts. More detailed information on the relationship are given in line 161 and line 414 onwards in the first version. In the revised version, we added some more literature and sentences (see response to comment lines 278-279).*

**Lines 13-14:** "the 21-days antecedent precipitation index" --> I'm not entirely versed in the conventional determination of the 21-day threshold. If it's based on specific constraints, might I suggest clarifying or possibly reevaluating its presentation in the text?
*The antecedent precipitation index (API) is an objectified measure of the soil water content based on the amount of precipitation that has fallen. The API is determined by the weighted sum of the previous daily precipitation values, whereby the weights decrease with the time elapsed decrease with time.*

*The DWD data catalogue used provides for both the 21-day and 30-day API. These are the most common models for modelling pre-moisture. As the natural hazards investigated in this study were examined in relation to heavy rainfall events occurring shortly before, the 21-day API is considered a useful parameter, as it reflects the medium-term conditions at the respective locations well and therefore an influence of this on the occurrence of sudden natural hazards was assumed (and proven in the analyses). We added some sentences for clarification in the Discussion (chapter 4.1).*

**Line 15:** "with no significant increase for gravitational mass movements" --> This finding seems to be at odds with existing literature. Could the authors confidently assert the robustness of their data in this regard? Further emphasis on this point might be warranted, especially given the indicated positive correlation between increasing rainfall and landslide occurrences in Figure 4.
*The limitation of the study results are adressed in the discussion section, especially in chapter 4.2. The data quality and availability is a major restriction. As recommended by RC3, more information on the data set (especially the damage database from DB) and its restrictions is also given in chapter 2.1.2 in the revised version.*

**Line 15:** "21 day threshold" --> Once again, the 21-day threshold's basis is not clear to me. If it's a discretionary choice, clarifying its rationale or considering its removal for a more straightforward explanation might be beneficial.
*See response to comment line 13-14.*

**Lines 15-16:** "The results underline the importance of gaining more precise knowledge about the impact of climate triggers on natural hazard-related disturbances" --> The connection to this conclusion in the abstract seems somewhat tenuous. A more detailed exposition in the text might help in directly leading the reader to this message. Alternatively, authors could consider revising the final message that summarizes their results.
*We find this conclusion locigal because, as the reviewer noted in the previous comments, it is surprising that only a small proportion of natural hazard processes on railroad lines can be linked to a heavy rainfall event. We would like to end the abstract with a more general sentence.*

**Lines 19-26:** "" --> This is a really nice start to the manuscript. It really caught my attention.
*Thank you for the comment.*

**Line 28:** "significant" --> I would advise against using 'significant' unless it is in a statistical context. Perhaps 'considerably' could serve as a suitable alternative.
*We agree that „considerable“ is more appropriate in this context as „significant“ and replaced it in the revised version.*

**Line 39:** "Within the framework of proactive natural hazard management" --> Is this framework well-documented in the literature, or is it an innovative concept proposed by the authors? A reference or a detailed explanation would be valuable for readers.
*There are several publications on this topic and different elaborations of natural hazard frameworks (e.g. Mühlhofer et al. 2023 (doi: 10.5772/55538). Most procedures are very similar and show steps such as exposure (localising the hazard), vulnerability, risk and risk defence.*

**Line 98:** "Starkregenindex SRI" --> Could the authors kindly provide a citation for this index? If it is a novel introduction, a detailed explanation within the manuscript would enhance understanding.

*We will include two references (Schmitt 2017 and Schmitt et al. 2018) in the revised version, as proposed by public comments from CC2.*

**Line 100:** "Lower Saxony" --> Given the frequent references to federal states, it may be helpful to include a reference map for those less familiar with Germany's geographical and political landscape.

*We reworked Figure 2 in the way that the location of the federal states mentioned in the manuscript is marked in map (a) to make it easier for readers outside Germany to identify the states.*

**Line 111:** "with event-specific search terms" --> Could you specify which search terms were used?

*We added some examples of search terms in the revised version (e.g. branch, tree, landslide, flood), but a complete list is unfortunately not available as data was provided in several years and by different colleagues from Deutsche Bahn.*

**Line 128:** "the period 1 January 2017-16 December 2020" --> I'm curious as to why data from earlier years and for 2021 were not included. This query also pertains to mass-wasting events. If they were absent in the data, do the authors know the reason behind.

*We used all data for the anaylsis which was available for us. The recording system at DB changed and has been raised to a new level of quality in 2017, so data from earlier years is not available for external use and not suitable. The data set for 2021 for tree fall events was different from the recording accurancy for the years 2017-2020. In order to avoid misinterpretation due to data inhomogenity, we decided to exclude the report from 2021 for the tree fall data set.*

**Line 132:** "Explanatory control variables" --> It would be immensely helpful to visualize these data. Is it possible to include them in one or more figures within the manuscript?

*As proposed by reviewer RC1, comments on dataset, point 5 we prepared an additional table of all used terms and variables in the revised version instead of a figure.*

**Lines 153-154:** "Thus, there are event locations where more than 50 heavy rainfall events from 2011 to 2021 can be found." --> Could this indicate that a single event, such as gravitational mass movements, might be attributed to multiple heavy rain events? This clarification would greatly aid in understanding the methodology applied.

*Yes, it is possible, that a single event might be attributed to multiple heavy rainfall events, but only if the rainfall events occur within three consecutive days. However, this is only the case in a low number of natural hazard events. There is only one tree fall event with two associated heavy rainfall events. There are also two gravitational mass movements, each with two associated heavy rainfall events, one on the day of the event and one the day before. There is no „double events" for floods. We added some explanatory sentences at the end of the following paragraph in the revised version.*

**Line 161:** "for deep landslides" --> It may be more accurate to refer to these as "deep-seated landslides" to ensure clarity and precision in terminology.

*We added deep-seated landlides to ensure clarity.*

**Line 166:** "the data set" --> It appears the term "dataset" is used inconsistently. Adopting a uniform usage could enhance the manuscript's readability.
*We replaced „data set" by „dataset" in the whole manuscript.*

**Line 270:** "events" --> Could you specify which events are being referred to here? Are these meant to be natural hazards?
*„Events" refer to „flooding events". We specified this in the text.*

**Lines 322-323:** "After two days, the probability of a gravitational mass movement is no longer different from a situation with no heavy rainfall."
*-> Unfortunately, there is no comment on this line. If there are any comments on this statement, we would be happy to discuss them.*

**Lines 325-326:** "In contrast, one day after a heavy rainfall event, a tree fall event is 2.4 times more likely to occur than in days with no heavy rainfall." --> The linkage of tree falls to heavy rainfall events, as highlighted here, is quite intriguing, especially considering that 98% of tree fall events were not directly linked to rainfall in the previous sections. Further clarification on this discrepancy would be beneficial.
*That's right. It is interesting that heavy rainfalls, although they only directly affect 2% of tree falls, can increase the probability of their occurrence. However, it must be said that - as written - it is only a relative increase to the case without an associated heavy rainfall event. Also, both events (heavy rainfall and tree falls) are very rare events in relation to the route network and the time period in days. This means that the probability of an event occurring is still low. We added these explanatory sentences in the revised version.*

**Lines 352–353:** "In the case of b) and e), the distance becomes greater at higher values, i.e., the higher the amount of accumulated precipitation, the more a heavy rainfall event increases the probability of occurrence of a flood or gravitational mass movement." --> It might be worthwhile for the authors to explore the findings of Saito et al. (2014; https://doi.org/10.1130/G35680.1), which suggest that increasing rainfall totals enhance landslide activity up to a certain threshold beyond which the effect does not apply. This could provide valuable insights into the study's analysis.
*Thank you for the reference to the paper by Saito et al. (2014). We included the reference and some explanatory sentences in the discussion (chapter 4.1) in the revised version.*

**Line 398:** "susceptible" --> The use of "susceptible" might be slightly misleading, potentially being confused with the term 'landslide susceptibility.' Perhaps "prone" could be a more precise alternative, avoiding any ambiguity.
*We replaced „susceptible" by „prone" in the revised version.*

**Lines 414-415:** "for gravitational mass movements it was as much as 17 % (Figure 1)" --> The low percentage of gravitational mass movements attributed to heavy rainfall seems somewhat unexpected, given that rainfall or related floods are the main triggers for landslides in Germany. This observation suggests the 2-day threshold might be too restrictive. Extending this threshold could potentially reveal a more nuanced impact of

rainfall on landslide occurrence. It's commonly suggested that lag times of 10 to 15 days are realistic, with some extreme cases even longer, which could be an important consideration for the study.
*See response to reviewer RC1, comments on Methods, point 2.*

**Figures:**
**Figure 1** --> The resemblance between subplot d and f is quite notable. This observation raises an interesting question: might the authors consider the possibility that landslides along tracks are more influenced by flood processes than by rainfall alone? This could potentially be attributed to processes such as slope undercutting or toe removal by floodwaters. In times these floods might be caused by a rain event upstream that is spatially not overlapping with the landslide location. Further exploration of this hypothesis could provide valuable insights.
*The good resemblance between subplot d and f is due to the years 2013 and 2016 where very rainy days/weeks occurred in May/June leading to a high number of heavy rainfall events and recorded natural hazards along the German railway network. However, a spatial and temporal intersection of two processes could only observed for a few cases in the datasets. We added some explanatory sentences in the figure description section in chapter 2.1.1.*

**Figure 2** --> It would be beneficial if the authors could provide a more detailed explanation of how the intersection depicted in this figure was calculated. This aspect seems crucial and might be more appropriately discussed in the results section to enhance the reader's understanding of the methodology. If it is discussed later than the figure's location, it might be an option to add a sentence summary in the caption.
*See response to reviewer RC1, comments on Datasets, point 1. We included a description on how the intersection was done in the revised version prior to the figure's location.*

**Figure 3** --> Currently, I find this figure somewhat challenging to interpret. Might the authors consider revising it to enhance its clarity and accessibility? Providing additional context or simplifying the presentation could improve its comprehensibility and overall impact on the reader's understanding.
*In fact, we would like to leave the illustration as it is, as we cannot find a simpler representation for this complex issue. The figure actually only shows the occurrence of all three natural hazards in total (left bar), with (middle bar) and without (right bar) associated heavy rainfall events by season (colours).*

**Reply on RC3**

**Major comments**

1. For the reader it is not easy to gain an impression of what the DB Netz AG database looks like. Because some of the results are somewhat puzzling and surprising (see point 2), we request the authors to provide more information on how this dataset looks. A few concrete suggestions:
    1. Include a few example records from the database showing the raw data (i.e. the raw text record that was identified), preferably with examples showing all three types of

natural hazards; and examples that did match with rainfall events and examples that did not match.

*The authors are not part of the DB Group, and the DB Netz AG database is an internal documentation that is not publicly available. Unfortunately, we have no rights to publish raw data from the database (it includes any disruption within the railway system) and only receive extracts from the database (as .csv) and have no access to the database itself. We are therefore unfortunately unable to show raw data for data protection reasons. In the revised version, we added some more descriptions on the dataset (chapters 2.1.2 and 4.2) and included a new figure showing one example for each type of natural hazard with a picture and the information given on the example event in the database. As the information on the processes itself are rather limited, the event information does not differ between events that match with rainfall events and those which not match. Therefore we show only one example per natural hazard process. We decided not to explicitly mention again the percentages of matches vs. non matches as these numbers are already mentioned as major results in the manuscript (e.g. lines 10-11).*

2. Please give an example(s) of how a well-known flood event (example the July 2021 floods) was mapped in the data, ideally with some pictures of how the damage looked in the real world. Earlier NHESS articles (e.g. https://doi.org/10.5194/nhess-22-3831-2022) and the author's https://doi.org/10.3390/atmos13071118 contain many relevant details that can be used to link the records to.

*The studies mentioned use different data sources than we use in this manuscript. Depending on the size, severity and impact of an event, there is an obligation to document it in varying degrees of detail, meaning that significantly more (and different) data is available for the July 21 event than for smaller events. The database used in our paper contains damage reports that are recorded during ongoing operations. This means that if traffic is completely suspended in the event of a major incident, the effects on the routes are only recorded at a later date and in a different form. In order to ensure comparability between the individual events and the years, we have decided for this manuscript to use only one data source for damage events and not to compile information from different sources. To give an impression how the damage look like, we used a picture from the July 21 event in the new figure (see response to previous comment).*

3. Please give examples of that were tagged as a flood by DB, but for which no extreme rainfall was reported.

*See responses to comments 2.1-2.4. We adressed this issue in more detail in the revised version in the discussion (chapter 4.2).*

2. We find the finding that only a quarter of the flood events could be linked to extreme rainfall events (line 414) very puzzling. In our view, the potential causes of this are insufficiently discussed in the draft version. Please critically reflect on the following possible causes:
   1. Is it possible that most of the reported flood events relate to river (fluvial) flooding? On the one hand, one might expect that this is the case, because it would present a natural explanation for the fact that the damage is observed in a different location than the extreme rainfall event. On the other hand, we would be surprised if the ratio between fluvial and pluvial flood events would be 3:1. Also, if 75% of the flood cases would concern river flooding, one may wonder if the methodology of the paper

is sound. The authors mention this aspect in line 463, but what is missing is a reflection on whether this explains the low correlation in their results. *As mentioned above, the information in the database on each event are limited, so that it is not possible to evaluate which proportion of the events relate to river flooding. Another limitation is that the spatial intersection does not take into account the whole catchment area and flow direction (see also comment from RC2, line 79). We added some explanatory sentences in the discussion (chapter 4.2). To summarize, however, we would agree that the linkage between flood events and heavy rainfall events would have been higher. As mentioned, the data unfortunately does not allow for a more in-depth analysis at the moment. Currently, the infrastructure operators (DB) are changing their documentation system for such events, so we hope, in the future, it will also be possible to compare events with images and unique keywords. Unfortunately, this is not relevant for the manuscript, as the changeover and generated data sets will only be available in a few years' time.*

2. Another take would be that most of the floods are caused by rainfall events that do not qualify as heavy rainfall, which is reasonable given that there is no standardized guideline for defining heavy rainfall (line 72). If this is the case, it would be best to clarify it in the text and highlight it as relevant further research. *As mentioned in the response to comment 2.1 above, the data quality of the data base from DB is limited and the whole catchment area is not taken into account. Precipitation events not classified as heavy rainfall events in the CatRaRE-data set but occurring in the catchment area could be detected by a detailed analysis of the HYRAS data set used in our study. However, that would be a very complex analysis (high resolution hydrological modelling over whole Germany), which goes beyond the scope of the study.*
3. Is it possible that due to other reasons, there is a mismatch between what the DB understands as a 'flood' is very different from what the radar data shows? *This may also be due to the fact that the database is not filled with events by experts, but by the staff on site along the route. It is therefore possible, that technical terms are not always used correctly and flood events are not differentiated according to their cause. As mentioned above, an event is only recorded if it disrupts the railway operations, so there may occur heavy rainfall events or river floods that are not captured in the dataset. We added some explanatory sentences in the discussion (chapter 4.2).*
4. Are there other possible explanations? *As already mentioned in the response to major comment 1, events are recorded in varying degrees of detail depending on their size and extent. In the case of line closures, there may be delays between the time of the event and the documentation of the event in the database. It must also be taken into account that only events that have caused disruption or damage to operations or infrastructure are recorded in the database. These points are mentioned in more details now in the discussion (chapter 4.2).*

We mainly raise the above points because for modelling studies, the outcomes of the present study may have large implications. Most modelling studies assume a deterministic relationship between rainfall, inundation and damage. The present study seems to suggest that such relationships would only explain 25% of reported flood damage events. We invite the authors to further reflect on this. Are the author's aware of any other empirical studies that looked into this relation? Did they find

similar results? *As mentioned in lines 487-489, there are only few scientific studies available on analysing natural hazard-related disruptions in the transport sector based on event databases from infrastructure operators. To our knowledge, none of them used a similar approach as presented in this manuscript.*

**Minor comments**

- In the abstract, please list the 'three associated natural hazards' upfront. Now it takes a while before the reader knows which hazards you examined, namely: floods, gravitational mass movements / landslides?, tree fall. *We added the three associated natural hazards in line 7.*
- Line 47: and smaller tolerance of risk compared to road transport? *Disturbances on railway lines have often more impact on the traffic than on roads because of fewer alternative routes and usually longer times for restorations. A lower risk tolerance therefore makes sense from the operator's perspective.*
- Line 50: Can the authors provide any additional information on what type of damage is reported in the DB database? Do railtrack characteristics play a role? What part of the track is damaged? Could it also be damage to a pier or abutment of a bridge that supports the track? *DB does not currently have guidelines on the specific information that must be recorded when recording incidents. The level of detail is therefore dependent on the person making the report. This means that for some events there is also information on which part of the rail system is damaged (e.g. overhead line, switch), but in most cases this remains unclear. Railtrack characteristics play a role particularly in the effects, e.g. the duration of the disruption. If, for example, the overhead line is damaged by a falling tree, it usually takes much longer to restore access to the line than if such an event occurs on a non-electrified line. We added some explanatory sentences in chapter 2.1.2.*
- Line 59: do you mean bias or correlation? *We mean correlation.*
- Line 64: One or two figures to illustrate the data sources could be useful for a reader that is unfamiliar with them. *We added a new Figure in the revised version of the manuscript (see response to Major comment 1.1).*
- Line 88: Figure 1 description indicates Monthly and yearly distribution of heavy rainfall, stating it as Yearly and monthly would better match the content of the figure. Figures c-h are also more likely to be consulted by the reader when reading the results, making the figure placement inconvenient (though understandable). Could changing the position or splitting the figure into two make it more readable? *Figure caption can be changed as proposed. The position of the figure can also be changed if wished by the editors. We do not find the division into two figures helpful, as in our opinion it is clearer if the monthly and annual distribution can be viewed directly next to each other.*
- Line 147, can you describe in a few lines how the polygon data looks like? Is it a polygon indicating a uniform amount of rainfall within that polygon? *The polygon area describes the entire area that has been issued as a heavy rainfall event by the DWD. The attributes of the polygon are standardised to the entire polygon. We added this information in the revised version.*
- Line 149: Punctuation can be improved for readability. *We rewrote the sentence for better readability.*
- Line 163: The selection of 2 days as a time window could be better explained; how often does it take longer than 2 days to record the damage events? During periods of heavy rainfall, it may not be safe to collect the data, for example? *Unfortunately, there are no*

*official or general statements about a possible delay. However, the documented events are all events that hinder and disrupt operations or infrastructure. This also means that in case of doubt not all events are documented, but only those that directly affect the rail in operation. Because of this, the time delay can be viewed as not very large (usually a maximum of a few hours) and negligible. For justification of the time window, we explained the natural breaks in the dataset in more detail in the revised version (see response to comment from RC1, Methods, comment 2).*

- Line 185-200: The definition of route segments could be further clarified and justified, specifically: (1) what is an "operating point"? (2) What are the implications of such wide range of lengths (140 m to 12.7 km) – is the starting point in a long stretch equally representative as that of a short one? (3) Taking 5-meter segments may be unmanageable, but why not, for example, use 1 km segments as a standard? *(1) An operating point is a railway system defined according to the Railway Construction and Operating Regulations (EBO). Most operating points fall into the categories of stopping point, block point or switch. Operating locations are an important measure and category in the railway industry and were therefore selected. (2) As operating points are not evenly distributed over the whole railway network, the lengths differ. This may lead to mis-representations, which are unavoidable, especially in combination with the different accuracies available for the geolocation of the natural hazard events. (3) The reported natural hazard events refer to route segments and a corresponding operation point, additional details vary. A geolocalisation on geometries or points that differ from this reference frame are difficult and prone to error. We included the definitions in the revised version.*
- Line 209 (and other location): I find the term natural hazards a bit ambivalent in this context, because it could be used to indicate either the extreme rainfall event, or the flood/gravitation mass movement/ tree fall. *The term is derived from its usage in the operational context, where the damage events due to natural hazards are separated from damage due to i.e. infrastructure failure. For a better orientation for the reader, we added a supplementary table (new Table 1) with definitions of the most important terms in the revised version.*
- Line 393: what is meant with: can be spatially overlaid. *We meant „spatially intersected" as shown in Figure 2. We replaced „overlaid" by „intersected".*
- Line 397: does this conclusion follow from the data, or from other literature, or from common understanding? *This conclusion is from common understanding, therefore no reference is added here. It is also supported from the damage data from DB, because events occur mainly on routes following valley courses or crossing low mountain ranges.*
- Line 401-402: Is flooding triggered by heavy rainfall events (Line 402) an example of the statement in the previous sentence? This can be clarified. The use of the word connections makes it sound like it is a separate idea or concept. *This is meant as an example for the statement in the previous sentence. We replaced the word „connections" by „relationships" for clarification.*
- Line 410: This idea is not very clear. An additional sentence with a specific example may help. Something like "[…] but to establish connections between the processes through X or Y, for example, by looking at mass land movements triggered by flooding", if this was indeed the idea. *We formulated an additonal sentence with a specific example in the style the reviewer proposed.*

- Line 451: What is meant by “the different background of the data collections”? Is it their intended or original purpose rather than their background? *„Intended or original purpose“ is correct; we replaced „background“ by this terminology.*

**Reply on CC2**

I have read this preprint with great interest and think that it is an important study for under-standing the influence of heavy rainfall events on damage to rail transport and infrastructure. My colleagues and I developed CatRaRE and we highly appreciate the use of the dataset in this study. However, there are a few issues in the description of CatRaRE and the results that I would ask the authors to address. In the case the authors have any questions or would like to discuss some of the issues mentioned below, please feel free to contact me or my colleague Ewelina Walawender.

- CatRaRE catalogue: In the abbreviation CatRaRE the word “catalogue” is already included, therefore CatRaRE catalogue would mean Catalogue of Radar-based heavy Rainfall Events catalogue. I suggest to use either just CatRaRE or catalogue of radar-based heavy rainfall events

*Thank you for the comment. We will use the term „CatRaRE“ in the revised version.*

- P.3, L.72: CatRaRE W3 and T5 are DOI referenced datasets. Please use the appro-priate reference for the catalogue used in this study, which I guess is the Version 2022.01:

Lengfeld, K., Walawender, E., Winterrath, T., Weigl, E., Becker, A., 2022, Heavy pre-cipitation events version 2022.01 exceeding DWD warning level 3 for severe weather based on RADKLIM-RW version 2017.002, DOI:10.5676/DWD/CatRaRE_W3_Eta_v2022.01.
In case another version is used, please check
https://www.dwd.de/DE/leistungen/catrare/catrare_daten.html?nn=16102&lsbId=751876

*We added the reference for the catalogue in the revised version. The version 2022.01 is correct.*

- P.3, L.73-74: In CatRaRE events with 11 different durations between 1 and 72 hours are listed. In the catalogue W3 we use the lower boundary of warning level 3 as a threshold. Not only the warning levels for 1 hour (25 mm) and 6 hours (35 mm) are used, but also the ones for 12 (40 mm), 24 (50 mm), 48 (60 mm) and 72 hours (90 mm). There are no official warning levels for rainfall events with durations of 2, 3, 4, 9 and 18 hours. Therefore, for these durations we linearly interpolate the official warning levels and get thresholds of 27 mm in 2 hours, 29 mm in 3 hours, 31 mm in 4 hours, 37.5 mm in 9 hours and 45 mm in 18 hours.

*Thanks for the clarification. We added a more detailed description in the revised version.*

- P.4, L.98 and Table 1: The SRI does not describe the speed at which rainfall accumulates within a specific duration of time. The SRI is based on the return period of the rainfall amount for indices 1-7, where 7 corresponds to a return period of 100 years. Indices 8-12 are based on the rainfall amount compared a precipitation with a return period of 100 years. Please clarify the description and see Schmitt (2017) and Schmitt et al. (2018) for more information.

*We added the correct definition and the references Schmitt (2017) and Schmitt et al. (2018) in the revised version.*

- P.6, L.136: What is the reason for choosing the HYRAS dataset over the climatological radar

dataset RADKLIM? RADKLIM would correspond to CatRaRE and has a higher spatial resolution comparable to the soil moisture dataset.

*The soil moisture and HYRAS dataset have the same resolution. It was chosen in connection with the soil moisture dataset, as they are both based on station based observations of precipitation.*

- P.10-11, L.252-256 + Section 3.3.: The CatRaRE event variables the authors have chosen (Tab. 1) are calculated for the whole event area (e.g. as an average over the event zone), that can in extreme cases cover several thousand km². However, the damage data used in this study are available for a given route segment (point location), so the cross-analysis makes sense only if a given rainfall event is undifferentiated within its zone in terms of precipitation characteristics (RR, SRI, V3) and occurs over an area with similar landscape pattern (TPI, VSGL, STRM). A pixel-based analy-sis would be more appropriate in this case. Also using the ETA as a measure of extremity is not proper in case of point-analysis, as it is calculated exactly on the basis of the event area.

*The accuracy of the geolocalisation of the damage events is fairly low. A pixel based evaluation would simulate a fake accuracy. Hence, we decided to use the area-averaged characteristics of the heavy rainfall event.*

- P.11, L.270: The fact that only 23% of the flooding events are linked to heavy rainfall events seems surprising to me. Did the authors also check for rainfall events in a certain radius around the flooding since rainfall does not necessarily cause flooding in the region of its occurrence but in the region where the water flows to.

*See response to comment from RC2, line 79.*

- P.13, L.312-314: It is not clear to me why a rain event should cause a flood one or two days after it's occurrence. If there is no more rain in the area there shouldn't occur a flood unless the water comes from another region, e.g. from upstream a river. But then the flood is probably triggered by another rainfall event that occurred upstream and not by the one that occurred in the area with the flooded railway section There-fore, not only a temporal but also a spatial buffer should be taken into consideration. In case the flooding occurred one or two days after the rainfall event I would also suggest checking the HYRAS dataset if there was more rainfall in the damaged area or the surroundings that could have caused the damage but wasn't classified as an event in CatRaRE. I understand that a detailed analysis of flow paths is beyond the scope of this paper, but the issue as well as the difference between damages caused by a heavy rainfall event and by a flood event should at least be mentioned in the discussion.

*We added the HYRAS dataset in our analysis with the intention to detect additonal rainfall events which are not included in CatRaRE. In our opinion, the approaches of the recording methods and the spatial reference from HYRAS and RADOLAN complement each other well. A detailed analysis of the flow path is not the focus of the paper, but we added some sentences on this issue in the discussion (chapter 4.2), as proposed by the reviewers.*

- Section 3.3: I am not sure if increasing each parameter by one unit is appropriate. 1 mm increase in mean precipitation is not comparable to increasing the duration by 1 hour or the SRI by 1. Let's e.g. assume a precipitation sum of 50 mm in 1 hour has a return period of 100 years, which corresponds to SRI = 7. Increasing the precipitation sum by 1 mm leads to 51 mm in 1 hours which most probably still has a SRI of 7 because the return period won't increase that much. Increasing the SRI by one to SRI = 8 would mean according to Schmitt (2017) that the precipitation would be 1.2 to

1.4 times the precipitation sum for return period of 100 years (which is 50 mm in our case). Therefore, increasing SRI from 7 to 8 would increase the precipitation sum from 50 mm to a value between 60 and 70 mm, which is 10 to 20 times more than the increase of 1 mm that was assumed for investigating the influence of increasing the precipitation by 1 unit. Therefore, the influence of increasing the SRI by one unit is by definition larger than the influence of increasing the mean precipitation by one unit. Also, I don't quite understand how the duration of precipitation is increased. In my example I had a duration of 1 hour and 50 mm. Does increasing the duration by one unit mean that it will rain 50 mm in 2 hours instead or 2*50 mm = 100 mm in 2 hours?
*Our analysis represents an initial evaluation of the quantitative effects of the control variables. Without further analysis, the values between the control variables are only comparable to a limited extent.*

**List of relevant changes made in the manuscript**

Figures and tables:

- One new figure (Figure 3) to visualize the damage database
- Two new tables (Table 1 and 2) for a better overview of used data and to give definitions of important terms

Chapter 2:

- More detailed descripition of the used datasets (damage database of DB InfraGO and CatRaRE)
- More detailed descripition of the statistical analysis and the used models with better description of the variables and addtional equations

Chapter 3:

- More detailed description of the statistical parameters and their interpretation

Chapter 4:

- More detailed discussion on other triggering factors than heavy rainfall events
- More process-specific discussion in particular for the delimitation of river floods versus flash floods and the influence of soil moisture on tree falls
- More detailed discussion on the limitation of the damage database

---

## Author Response (AR2)

**Response to Reviewer #1**

The authors made an extensive revision of the manuscript and have improved many aspects. Very helpful are the tables describing terminology and variables. Unfortunately, the most important weakness of the study (overfitting) was not addressed in the revised version. In my original review I asked if overfitting can be ruled out. The authors admitted that this is not the case but no consequences followed.

*Overfitting is addressed in the newly added Appendix A of the paper where we present results of simpler versions of the model and show that the magnitude of the results do not change if a simpler model is selected. However, the simpler models are severely prone to omitted variable bias. Therefore we have decided to keep the original model in the main text because they take other important confounding factors into account and are, in our view, the most appropriate for understanding the relationship between heavy rain and natural hazards along the rails.*

An easy to read publication about overfitting can be found under the following DOI: 10.1097/01.psy.0000127692.23278.a9
Citing from this publications:
1) "Taken to its extreme, if the number of unknowns in a model is equal to the number of observations, the model will always fit the sample data perfectly, even if all the predictors are noise, ie, entirely unrelated to the response variable."
2) "For linear models, such as multiple regression, a minimum of 10 to 15 observations per predictor variable will generally allow good estimates."
3) "In the case of models with a binary response, if the number of events is smaller than the number of nonevents, the limiting sample size is the number of events."
4) "If we cannot gather a sample of sufficient size, we have to find ways to simplify our model..."
*We have opted for reducing the complexity of the model (option 4) and we show in the appendix that the primary interpretations hold with a less-complex model.*

In the light of these statements it is absolutely essential that the authors perform tests to demonstrate that the results (e.g. the statistical significance values) are meaningful and not artefacts of overfitting. Such a test could be for example a cross validation. The prediction for the years not used for training should have more skill than forecasting the climatological probability of the respective season. The large number of extra coefficients associated with the random variable, alone, makes me wonder if there can be any skill in the statistical models - especially the ones for gravitational mass movements.

*We would like to emphasize that the main goal of the paper is to shed light on the relationship between heavy rain and the three natural hazard processes, and not to develop an accurate predictive model of natural hazards. Therefore we have determined that a cross validation procedure, which evaluates the models based on their predictive capacity, is not the appropriate approach to tackle the issue of overfitting for this analysis. Instead we have chosen, as mentioned, to present simpler models with different permutations of the control variables, and demonstrate that in cases that are at most risk of overfitting, selecting a simpler model will not alter the conclusions.*

In the revised manuscript the authors write for example: "Despite the lower number of observations of events available for the gravitational mass movements, the model and chosen variables describe the relationship between gravitational mass movements and heavy rainfall events accurately. In the case of tree fall events the actually by a magnitude higher number of data points available for the

calculation does not lead to a better model fit...."
I believe that the perfect fit for gravitational mass movements is an artefact of overfitting.
*For gravitational mass movements, the odds ratios of heavy rain at the mean values of the meteorological variables (which can be found at the bottom of Table A3, columns (4) – (6)) are similar in magnitude to the odds ratio of heavy rain in the simple logit regression in Table A3 column (1). If the simpler model were to be selected, the magnitude of the effect of heavy rain on gravitational mass movements would change little. Nevertheless, the loss of statistical significance in the coefficient of heavy rain in columns (4) to (6) of Table A3 makes it difficult to make a solid conclusion based on the current data alone. More data is required for more robust results.*

Something that should also be improved before publication is the description of the statistical models.
I still find it very confusing to understand what the final statistical model looks like. In the answer to the review the authors state: "For each hazard model, 5 coefficients and one random variable had to be fitted." This agrees with equation 2 as $\beta'_2$ includes 3 coefficients.
Later in the manuscript the authors write: "Annual and seasonal dummies are also included to account for the fact that the number of natural hazards varies greatly in different years and seasons." These additional variables don't show up in the equations and are not included in table 2. The hazard indication for slope and embankment landslides is also only mentioned in the text but missing in table 2 and the model equations and/or the vectors listing the control and random variables. With these extra variables the model becomes more complex and includes even more coefficients.
To allow the reader to understand the study, the full model equations and a complete table with all the variables is essential. The incomplete tables and equations make this part untransparent.
*We have adjusted the description of the statistical models in Section 2.2.3. to explicitly describe all included control variables and interaction terms. We have also adjusted the calculation of the odds ratio to reflect the fact that the odds ratio will vary depending on the values of the meteorological control variables due to the interaction terms. In light of this change in the calculation, we have added Table 3 to the results section presenting the calculated odds ratios at the mean and median values of the meteorological variables. In the Appendix, the full regression tables area also presented will all included variables, seasonal and year controls and interaction terms.*

Outline: In their reply to the reviews the authors explain why they mixed the contents of sections. I must admit I am not too happy with this decision. When reviewing the manuscript I found it difficult to find the relevant information when trying to retrace certain aspects. The data section includes results (the analysis of the seasonal variation of the combined hazard). The rail segments are provided by DB (and not work done for this study). Therefore they should be described in the data section and not in the methods section (L272-281). It is up to the editor to decide if the outline can stay as it is or if it needs to be revised.^
*There was no mention by the editor as to whether a change in the outline is necessary.*

Minor remark:
Equation 3: x needs to be replaces with epsilon
*The greek letter epsilon is not used in any of our equations.*

---

## Author Response (AR3)

**Response to Reviewer 1**

The revised version of the manuscript shows substantial improvement over the previous version. The methods section now includes a complete and transparent description of the statistical models. An appendix has been added that includes a sensitivity analysis comparing statistical models of different complexity. This helps to interpret the results in the main paper.

The appendix shows that some features of the statistical model are robust and not an artefact of overfitting (e.g. heavy precipitation increases the odds ratios, precipitation close to the event is more important than precipitation prior to the event). The absolute numbers of the odds ratios are however not always stable. In these cases the absolute values of the odds ratios should not be interpreted.

*We have removed the interpretation of specific numerical values of the odds ratios for the meteorological controls.*

I agree with the authors that a balance should be found between the omitted variable bias and overfitting. However, the sensitivity study in the appendix does not allow to decide if/at which stage of complexity overfitting is present/starts. That the most complex model has the lowest AIC is not sufficient to prove that the most complex model is not affected by overfitting. (an examples for this can be found e.g. here https://stats.stackexchange.com/questions/524258/why-does-the-akaike-information-criterion-aic-sometimes-favor-an-overfitted-mo). Some of the relationships between predicant and predictor shown in the manuscript are unexpected and contradict physical reasoning. In these cases I strongly assume that this is a consequence of overfitting. Examples for surprising results for which no physical explanation is offered in the discussion are:

*We cannot rule out overfitting for our results, especially for gravitational mass movements where we have only a few hundred data points. However, many of the succeeding points can be explained either through technical, operational or hydrological reasons.*

Line 329: "For gravitational mass movements, the coefficient is not statistically different from one, meaning there is no evidence of a statistically significant difference between the odds of a gravitational mass movement with and without heavy rainfall." The values in table 4 at day of heavy rainfall and 1 day after heavy rainfall are statistically significant. How can this be?

*This is due to the fact that the model in Table 2 contains interaction terms (equation (3.1)), and the model in Table 4 contains no interaction terms (equation (3.2)). We have refrained from including interaction terms in the second model to maintain simplicity, as interactions were not relevant for the question being tackled in the second model.*

Line 377: "For tree fall events, the odds ratio on the day of a heavy rainfall is 0.296 and statistically significant, meaning that the odds of a tree fall event occurring on the same day as a heavy rainfall is less than a third that of a situation when no heavy rainfall occurs. In contrast, one day after a heavy rainfall event, a tree fall event is 2.4 times more likely to occur than in days with no heavy rainfall." Is there any physical reason to explain this. I assume this result is an artefact caused by the fact that the complex model includes two predictors for the event day that have to "share" the odds ratios (day of heavy rain and precipitation at route segment (at the event day)).

*The explanation for this is an operational one and lies in the way how the data for tree fall is collectedby Deutsche Bahn. As mentioned in Secion 4.2, the data collection for the DB damage database is done solely for the purpose of ensuring the safe and swift resumption of railroad operations. Tree fall events are reported by train operators only upon encountering them en route. When a heavy storm or rainfall is expected, often train journeys are cancelled in advanced to ensure*

*the safety of passengers and employees. Therefore, less trains travel on days of heavy rain, making it less likely to encounter tree fall events on the same day. Only after the storm has settled does the DB deploy their workers to survey the damage, and then most of the tree fall disruptions along the rails are encountered and reported.*

Line 429: "For c) and i), the curve "with heavy rainfall event" has a U-shape. Thus, the probability of a natural hazard occurring during a heavy rainfall event is higher when the soil moisture takes on extreme values than when it takes on average values. In the case of tree fall, this is particularly the case for low soil moisture values, and in the case of floods for high soil moisture values. The arc shape in f) indicates that the probability of occurrence is highest at medium soil moisture values."
If there is a physical reason for the shape of the curves in c,f,i, please explain. Why do medium values show the opposite signal compared to both extremes? I think this relationship is not related to physics but probably caused by overfitting.
*For c): There is evidence that extremely dry and extremely wet soil are determinants of floods (Vichta et al. 2024), mainly due to the hydrophobic properties of soil, and oversaturation, respectively. Heavy rainfall in an environment with very dry and hydrophobic soil or very wet and oversaturated soil can therefore easily trigger a flood event, and this is reflected clearly in the results. In moderate soil moisture cases, where the soil can still absorb water brought about by heavy rains, the effect of heavy rain is then less pronounced.*
*For i): It has been shown that drought stress can cause tree mortality (Grote, et al. 2016), meanwhile, soil oversaturation can cause waterlogging stress in trees (Gill, 1970; Kreuzweiser & Rennenberg, 2014). With trees that are already under stress and vulnerable in very dry or very wet soil conditions, a heavy rainfall event could cause additional stress and be more likely to trigger tree fall.*
*These explanations have been added to the text of the paper.*
*For f): We have added in the paper that we refrain from interpreting this result because of the insignificance of the coefficient of heavy rain on gravitational mass movement.*

Line 452: "the mean elevation within the heavy rainfall area reduces the odds of gravitational mass movement events". More events at higher elevation might be interpreted as an indication for the existence of a hill (with a slope). I cannot think of any physical reason that explains less events at higher elevations.
*There are several possible aspects to this result. First the number of gravitational mass movement events is very small in the available data set, so more data is needed for a robust validation of this finding. Second, train lines in Germany are more often located in lower-elevated areas, and for train lines built in higher elevated areas, more attention is likely given to landslide prevention. However, a deeper analysis of the distribution of railway lines among different elevations, not to mention better (and more) gravitational mass movement data, is necesary to validate these possible explanations.*

I still think that unphysical predictors such as the year of the observation, lessen the informative value of the statistical model. As long as only the robust results are interpreted in the article, this is acceptable.
I therefore suggest that the authors add a sentence stating that they will only describe and interpret robust results. All other "results" should be removed from the manuscript. Then the manuscript can be published.8
*We have added in line 310 a sentence saying we will only interpret robust result*s.

Further comments:

Line 401: "using the results of the interaction terms" Is this based solely on the interaction term or on the full model including the interaction term? In case this for the interaction term only: The odds ratio for the interaction terms is not statistically significant in many of the combinations. In line 352 you argue that you don't discuss results which are not statistically significant. Here you even include a figure+discussion. I would remove figures and discussion for the not-significant combinations. In case this is for the full model: What are the values for the variables that are held constant (mean/median?)?

*We used the full model to generate the graphs in Figure 4. The figure is generated by using the predictive margins approach (Williams, 2012): for each observation in the dataset, the actual observed values of the control variables are used to calculate the predicted probability of a hazard event if there had been heavy rain and if there had been no heavy rain. These observations were then ordered according to relevant meteorological variables, say precipitation, and the predicted probabilities with and without heavy rain for observations with the same precipitation levels were averaged. Therefore, for each value of precipitation, we get two points: the average probability with heavy rain (points on the dashed line) and the average probability without heavy rain (points on the solid line).*

*We have added an short explanation of the approach in line 370.*

Line 490: is V3 really small scale?

*All variables in the CatRaRE data set are calculated for the event zone. As most of the heavy rainfall events are of small extent, we assumed that the data describes the local climate conditions. For better understanding, we replaced the formulation „small scale" by „local".*

---

## Author Response (AR4)

**Response to Editor comments**

For publication, I ask you to make the following technical corrections:

In the responses to the reviewer comments, you provide explanations regarding lines 377 and 452. Please insert shortened versions of these explanations into your manuscript.

*We inserted shortend versions of the explanation in our manuscript.*

Table 2.1. should certainly read Table 3. Please check if the references to table 3 and 4 are still correct.

*We relabeled Table 2.1 as Table 3 in the table captions and checked for the correctness of references to Table 3 and 4.*